# A double dissociation between semantic and spatial cognition in visual to default network pathways

Tirso RJ Gonzalez Alam[1,2,3]*, Katya Krieger-Redwood[1,2], Dominika Varga[4], Zhiyao Gao[5], Aidan J Horner[1,2], Tom Hartley[1,2], Michel Thiebaut de Schotten[6,7], Magdalena Sliwinska[8], David Pitcher[1,2], Daniel S Margulies[9], Jonathan Smallwood[10], Elizabeth Jefferies[1,2]

[1]Department of Psychology, University of York, North Yorkshire, United Kingdom; [2]York Neuroimaging Centre, Innovation Way, Heslington, North Yorkshire, United Kingdom; [3]School of Human and Behavioural Sciences, Bangor University, Gwynedd, Wales, UK, York, United Kingdom; [4]Sussex Neuroscience, School of Psychology, University of Sussex, Brighton and Hove, United States; [5]Department of Psychiatry and Behavioral Sciences, Stanford University School of Medicine Stanford, Stanford, United Kingdom; [6]University of Bordeaux, CNRS, CEA, IMN, Bordeaux, France; [7]Brain Connectivity and Behaviour Laboratory, Sorbonne Universities, Paris, France; [8]Department of Psychology, Liverpool John Moores University, Liverpool, United Kingdom; [9]Integrative Neuroscience and Cognition Center (UMR 8002), Centre National de la Recherche Scientifique (CNRS) and Université de Paris, Paris, France; [10]Department of Psychology, Queen's University, Kingston, Kingston, Canada

*For correspondence:
t.gonzalezalam@bangor.ac.uk

Competing interest: The authors declare that no competing interests exist.

## eLife assessment

This **useful** experiment seeks to better understand how memory interacts with incoming visual information to effectively guide human behavior. Using several methods, the authors identify two distinct pathways relating visual processing to the default mode network: one that emphasizes semantic cognition, and the other, spatial cognition. The evidence presented is **solid** and will be of interest to cognitive and systems neuroscientists.

**Abstract** Processing pathways between sensory and default mode network (DMN) regions support recognition, navigation, and memory but their organisation is not well understood. We show that functional subdivisions of visual cortex and DMN sit at opposing ends of parallel streams of information processing that support visually mediated semantic and spatial cognition, providing convergent evidence from univariate and multivariate task responses, intrinsic functional and structural connectivity. Participants learned virtual environments consisting of buildings populated with objects, drawn from either a single semantic category or multiple categories. Later, they made semantic and spatial context decisions about these objects and buildings during functional magnetic resonance imaging. A lateral ventral occipital to fronto-temporal DMN pathway was primarily engaged by semantic judgements, while a medial visual to medial temporal DMN pathway supported spatial context judgements. These pathways had distinctive locations in functional connectivity space: the semantic pathway was both further from unimodal systems and more balanced between visual and auditory-motor regions compared with the spatial pathway. When semantic and spatial context information could be integrated (in buildings containing objects from a single category), regions at the intersection of these pathways responded, suggesting that parallel

processing streams interact at multiple levels of the cortical hierarchy to produce coherent memory-guided cognition.

## Introduction

The default mode network (DMN) is involved in higher-order cognition including in semantic cognition, mental time travel, and scene construction (*Andrews-Hanna et al., 2010b*; *Andrews-Hanna et al., 2010a*; *Ralph et al., 2017*; *Spreng et al., 2009*). Its functions and architecture are plagued by apparent contradictions: it often deactivates in response to visual inputs yet it is connected to visual cortex (*Knapen, 2021*; *Leech et al., 2012*; *Szinte and Knapen, 2020*). In addition, this network is associated with both abstraction from sensory-motor features (; *Chiou et al., 2019*; *Gonzalez Alam et al., 2021*; *Rice et al., 2015b*; *Rice et al., 2015a*) and internally generated states like imagery and autobiographical memory (*Philippi et al., 2015*; *Ritchey and Cooper, 2020*; *Spreng and Grady, 2010*; *Zhang et al., 2022*). A recent perspective suggests these diverse functions are facilitated by the topographical location of DMN on the cortical mantle (*Smallwood et al., 2021*). DMN is maximally separated from sensory-motor regions – both in terms of its physical location and in connectivity space. It is at one end of the principal gradient of intrinsic connectivity that captures the separation of unimodal and heteromodal cortex (*Margulies et al., 2016*) and this location is thought to allow DMN to sustain representations that are distinct from sensory-motor features and at odds with the current state of the external world (*Murphy et al., 2019*; *Murphy et al., 2018*).

Despite these common functional characteristics of DMN, parcellations of intrinsic connectivity reveal subdivisions (*Andrews-Hanna et al., 2010b*; *Schaefer et al., 2018*; *Wen et al., 2020*; *Yeo et al., 2011*). Lateral fronto-temporal (FT) DMN regions are associated with semantic cognition, including the abstraction of heteromodal meanings from sensory-motor features and the ability to access these meanings from sensory inputs in a task-appropriate way (; *Chiou et al., 2019*; *Ralph et al., 2017*; *Wang et al., 2020*). In contrast, scene construction, thought to be a key component of episodic recollection, is associated with a medial temporal (MT) subsystem (*Andrews-Hanna et al., 2010b*; *D'Argembeau et al., 2010*; *Hassabis et al., 2007*; *Zhang et al., 2022*). FT and MT-DMN subnetworks are interdigitated in regions of core DMN (*Braga and Buckner, 2017*) and they are assumed to work together but little is known about how information within them is integrated. One hypothesis is that the spatial adjacency of DMN subsystems allows their common recruitment and coordination when semantic and scene-based information is aligned; e.g., when semantically similar objects are found in a common location, or spatial position predicts the meanings of items that are found there.

FT and MT-DMN support heteromodal representations and yet can be accessed by visual inputs, raising the question of how neural pathways between vision and DMN are organised. Visual neuroscience has revealed different responses associated with recognising objects and scenes (*Kravitz et al., 2013*; *Kravitz et al., 2011*). Objects engage a ventral pathway extending laterally and anteriorly through ventral lateral occipital cortex (LOC) and the fusiform gyrus towards the anterior temporal lobes (ATL), thought to be a key heteromodal hub for conceptual representation. This pathway might act as input to FT-DMN (*Andrews-Hanna et al., 2014*; *Andrews-Hanna and Grilli, 2021*; *DiCarlo et al., 2012*; *Kravitz et al., 2013*; *Malach et al., 2002*). Navigating visuospatial environments and scene construction, on the other hand, involves the occipital place area, posterior cingulate, retrosplenial, entorhinal, and parahippocampal cortex, before this pathway terminates in hippocampus. These regions are associated with the MT-DMN subnetwork (*Andrews-Hanna et al., 2014*; *Andrews-Hanna and Grilli, 2021*; *Epstein and Baker, 2019*; *Kravitz et al., 2011*; *Reagh and Yassa, 2014*). This work suggests that visual and DMN subsystems may be linked. For example, during memory for people and places, medial parietal cortex mirrors the well-established medial-lateral organisation of ventral temporal cortex during the perception of scenes and faces; medial parietal regions also show differential connectivity to these visual regions (*Margulies et al., 2009*; *Silson et al., 2019*; *Steel et al., 2021*). Yet these past studies did not examine whole-brain connectivity or semantic cognition beyond the social domain and were also unable to explore the interaction of these pathways.

Here, we used multiple neuroscientific methods to delineate the pathways from visual cortex to DMN, providing convergent evidence for two parallel streams supporting semantic and spatial cognition. In Study 1, participants learned about virtual environments (buildings) populated with

objects belonging to diverse semantic categories, both man-made (tools, musical instruments, sports equipment) and natural (land animals, marine animals, birds). We then used functional MRI (fMRI) to examine neural activity as participants viewed object and scene probes and made semantic and spatial context decisions. Some buildings were associated with a specific semantic category (e.g. a building filled with musical instruments), while others included a mix of categories, allowing us to examine the interaction between semantic and spatial cognition. We identified dissociable pathways of connectivity between different parts of visual cortex and DMN subsystems; these overlapped with visual localiser responses for objects and scenes (in Study 2), as well as previously described DMN subsystems, and showed different patterns of functional and structural connectivity (in Study 3). These pathways refer to regions that are coupled, functionally or structurally, together, providing the potential for communication between them. They also had distinctive locations in a functional space defined using whole-brain gradients of connectivity: the semantic pathway was further from unimodal systems and more balanced between visual and auditory-motor regions compared with the spatial pathway. Moreover, when semantic and spatial context information could be combined (e.g. when the objects in a building were from the same semantic category), regions at the intersection of these pathways responded, in both DMN and visual cortex, suggesting these parallel processing streams can interact at multiple levels of the cortical hierarchy to produce coherent memory-guided cognition.

## Results

### Behavioural results

To examine task accuracy, we performed a 2×2 repeated-measures ANOVA using task (2 levels: semantic, spatial context) and condition (2 levels: mixed-category building [MCB], and same-category building [SCB]) as factors. There was a main effect of task ($F_{(1,26)}=76.52$, $p<0.001$), condition ($F_{(1,26)}=11.31$, $p=0.002$) and a task by condition interaction ($F_{(1,26)}=14.51$, $p<0.001$). Participants showed poorer accuracy in the spatial context task relative to the semantic task and in the MCB relative to the SCB condition. Participants were significantly less accurate in the MCB trials relative to the SCB trials of the spatial context task ($t(26)=4.08$, $p<0.001$); this difference was not observed in the semantic task ($t(26)=0.74$, $p=0.47$).

Response times showed the same pattern, with main effects of task ($F_{(1,26)}=51.37$, $p<0.001$), condition ($F_{(1,26)}=31.14$, $p<0.001$), and their interaction ($F_{(1,26)}=29.48$, $p<0.001$). Participants had slower reaction times in the spatial context task than the semantic task and in the MCB relative to the SCB condition. Post hoc comparisons confirmed that participants were significantly slower in MCB than SCB trials of the spatial context task ($t(26)=6.08$, $p<0.001$), but this difference was not observed in the semantic task ($t(26)=0.1$, $p=0.92$). These results are shown in *Figure 1*.

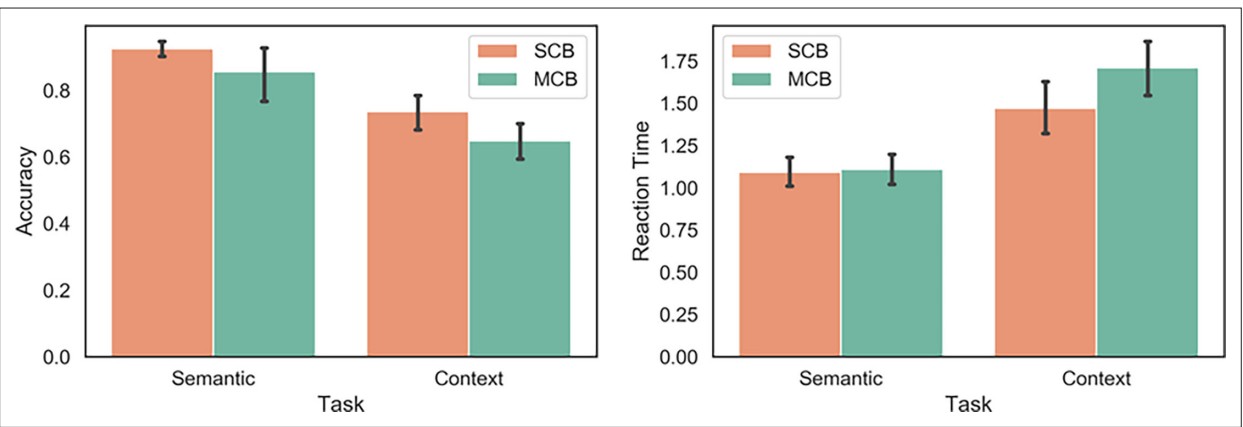

**Figure 1.** Behavioural results for the semantic and spatial context tasks inside the scanner. SCB = same-category buildings: all the items in the building were taken from the same semantic category. MCB = mixed-category buildings: the items in the buildings were drawn from different semantic categories.

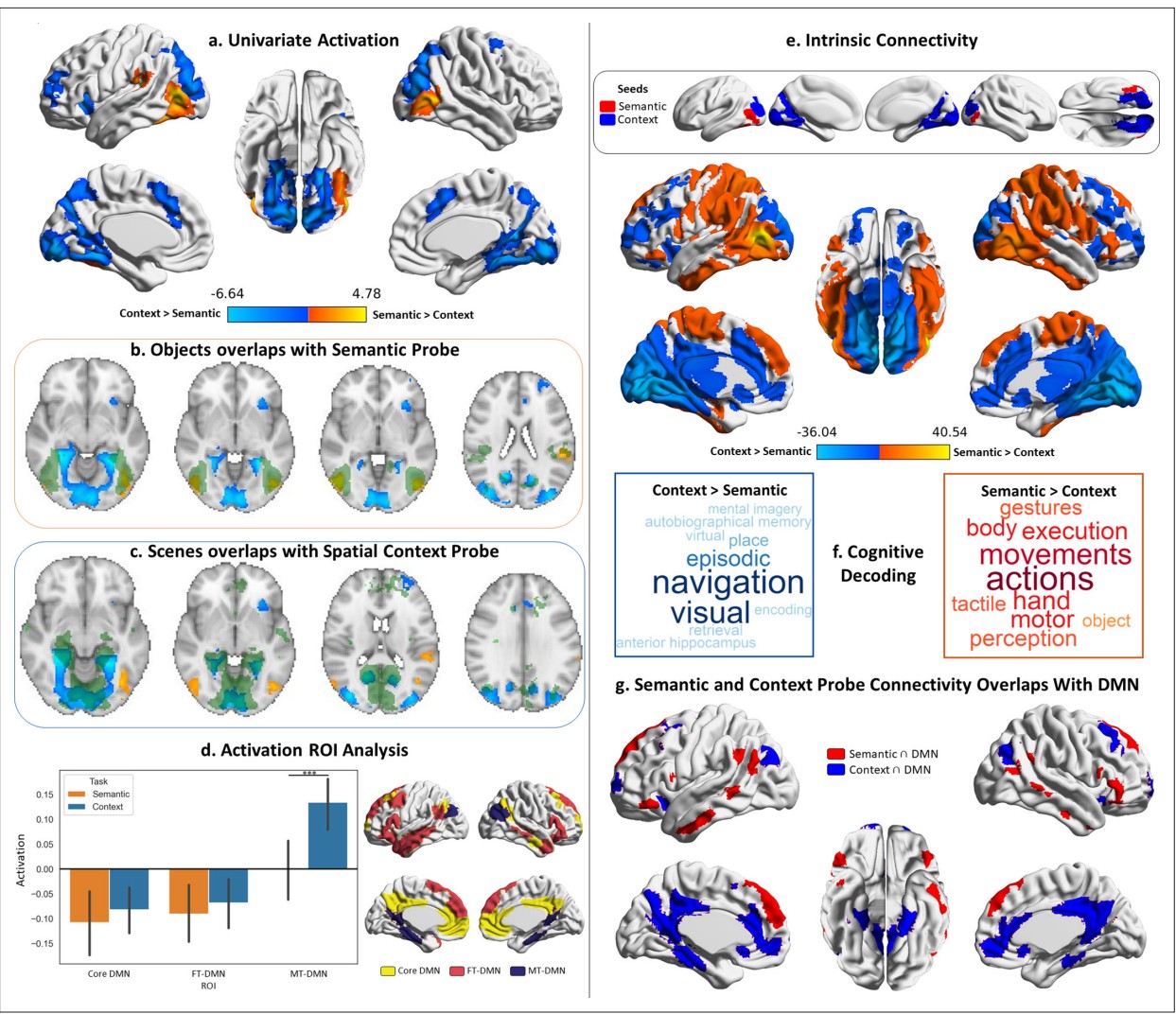

**Figure 2.** Probe responses. Warm colours = semantic > spatial context probes. Cool colours = spatial context > semantic probes. Left panel: Univariate results from Study 1, contrasting semantic and spatial context probes. Right panel: Intrinsic connectivity results from Study 3 using semantic and spatial context probe activation within visual networks as seeds. (**a**) Brain maps depicting the suprathreshold univariate activation results for the probe phase of the semantic and spatial context tasks. (**b and c**) Axial slices showing the overlap of these univariate results with scene and object localiser maps from Study 2 (the localiser maps are in green, and the univariate results maps are in warm and cool colours; the localiser maps are shown in *Figure 2—figure supplement 1*). (**d**) Region of interest (ROI) analysis examining the activation in the three default mode subnetworks of the Yeo 17 parcellation during the probe phase of the semantic and spatial context tasks. The error bars in the bar plots depict the standard error of the mean (Note: ***p<0.001); the ROIs are shown to the right of the bar plots. (**e**) Brain maps depicting the seeds and intrinsic connectivity results for the semantic and spatial context probe regions. (**f**) Word clouds depicting the cognitive decoding of unthresholded connectivity maps for semantic and spatial context probe seeds using Neurosynth (bigger words reflect stronger correlation of the functional maps with the terms); the colour code follows that of the brain maps. (**g**) Brain maps showing the overlap of these intrinsic connectivity maps for semantic and spatial context probes with the default mode network from the 7-network parcellation from *Yeo et al., 2011*.

The online version of this article includes the following figure supplement(s) for figure 2:

**Figure supplement 1.** Left panel: Areas associated with the passive viewing of objects are shown in red, and those associated with the passive viewing of scenes are shown in blue; areas that responded to both objects and scenes are shown in purple.

## Neuroimaging results

To probe the organisation of streams of information between visual cortex and DMN, our neuro-imaging analysis strategy consisted of a combination of task-based and connectivity approaches. We first delineated the regions in visual cortex that are engaged by the viewing of probes during our task (*Figure 2*), as well as the DMN regions that respond when making decisions about those probes (*Figure 3*): we characterised both by comparing the activation maps with well-established

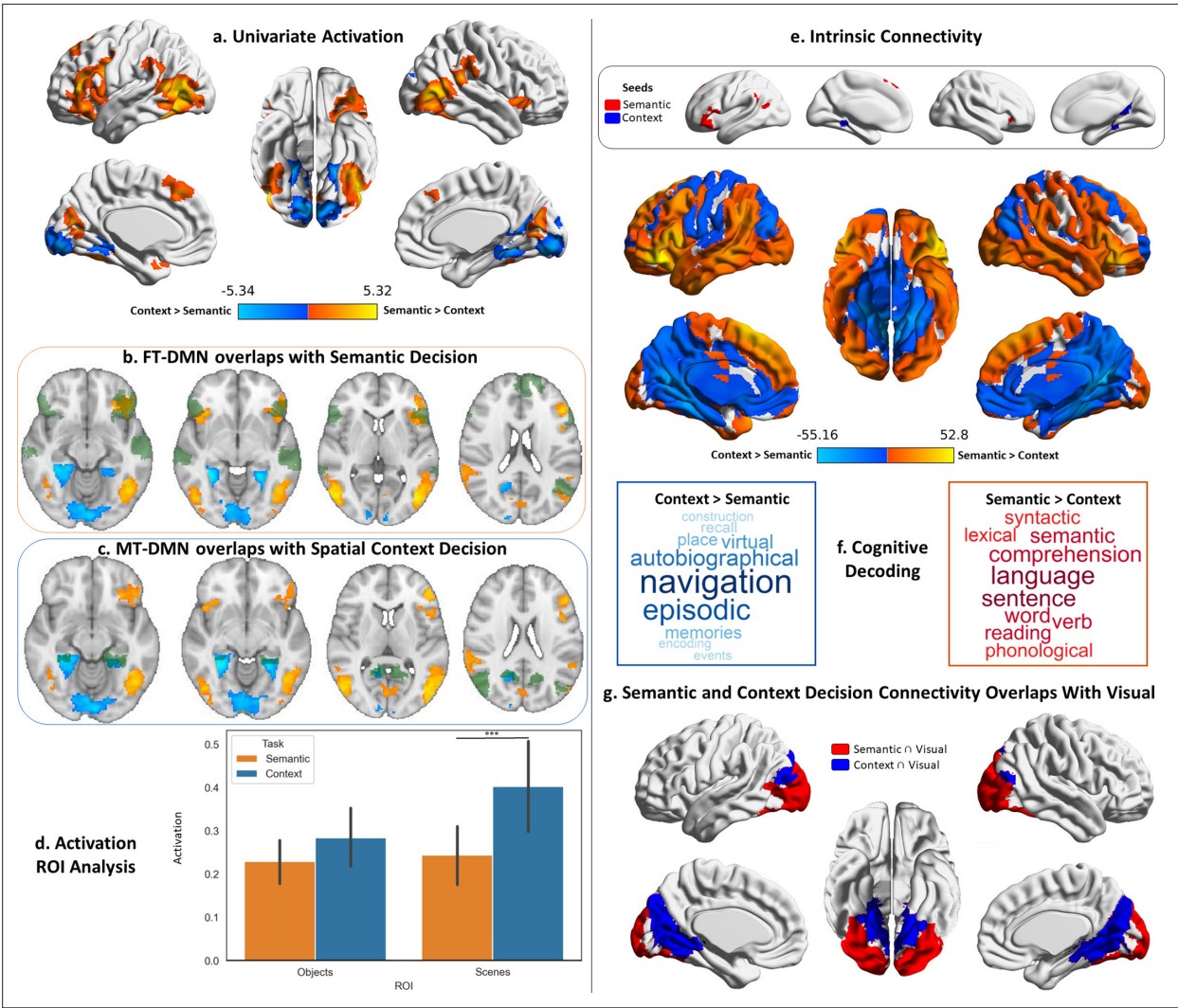

**Figure 3.** Decision responses. Warm colours = semantic > spatial context decisions. Cool colours = spatial context > semantic decisions. Left panel: Univariate results from Study 1 contrasting semantic and spatial context decisions. Right panel: Intrinsic connectivity results from Study 3 using semantic and spatial context decision activation within default mode network (DMN) as seeds. (**a**) Brain maps depicting the suprathreshold univariate activation results for the decision phase of the semantic and spatial context tasks. (**b and c**) Axial slices showing the overlap of these univariate results with the fronto-temporal (FT) and medial temporal (MT) default mode subnetworks of the Yeo's 17-network parcellation (the default mode maps are in green, and the univariate results maps are in warm and cool colours). (**d**) Region of interest (ROI) analysis examining the activation in the scene and object localiser maps from Study 2 during the decision phase of the semantic and spatial context tasks. The error bars in the bar plots depict the standard error of the mean (Note: ***p<0.001, *p<0.05); the ROIs are shown in *Figure 2—figure supplement 1*. (**e**) Brain maps depicting the seeds and intrinsic connectivity results for the semantic and spatial context decision regions. (**f**) Word clouds depicting the cognitive decoding of unthresholded connectivity maps for semantic and spatial context decision seeds using Neurosynth (bigger words reflect stronger correlation of the functional maps with the terms); the colour code follows that of the brain maps. (**g**) Brain maps showing the overlap of these intrinsic connectivity maps with the visual network from the 7-network parcellation from *Yeo et al., 2011*.

DMN and object/scene perception regions, analysed the pattern of activation within them, their functional connectivity and task associations. Having characterised these dissociable visual and DMN regions, we proceeded to ask whether they are differentially linked: are the visual regions activated by object probe perception more strongly linked to DMN regions that are activated when making semantic decisions about object probes, relative to other DMN regions? Is the same true for visual regions associated with scene perception and DMN regions responding to spatial decisions about which rooms were in the same building? We answered this question through a series of connectivity analyses (*Figure 4*) that examined: (1) if the functional connectivity of visual-to-DMN regions (and DMN-to-visual regions) shows a dissociation, suggesting there are object semantic and spatial

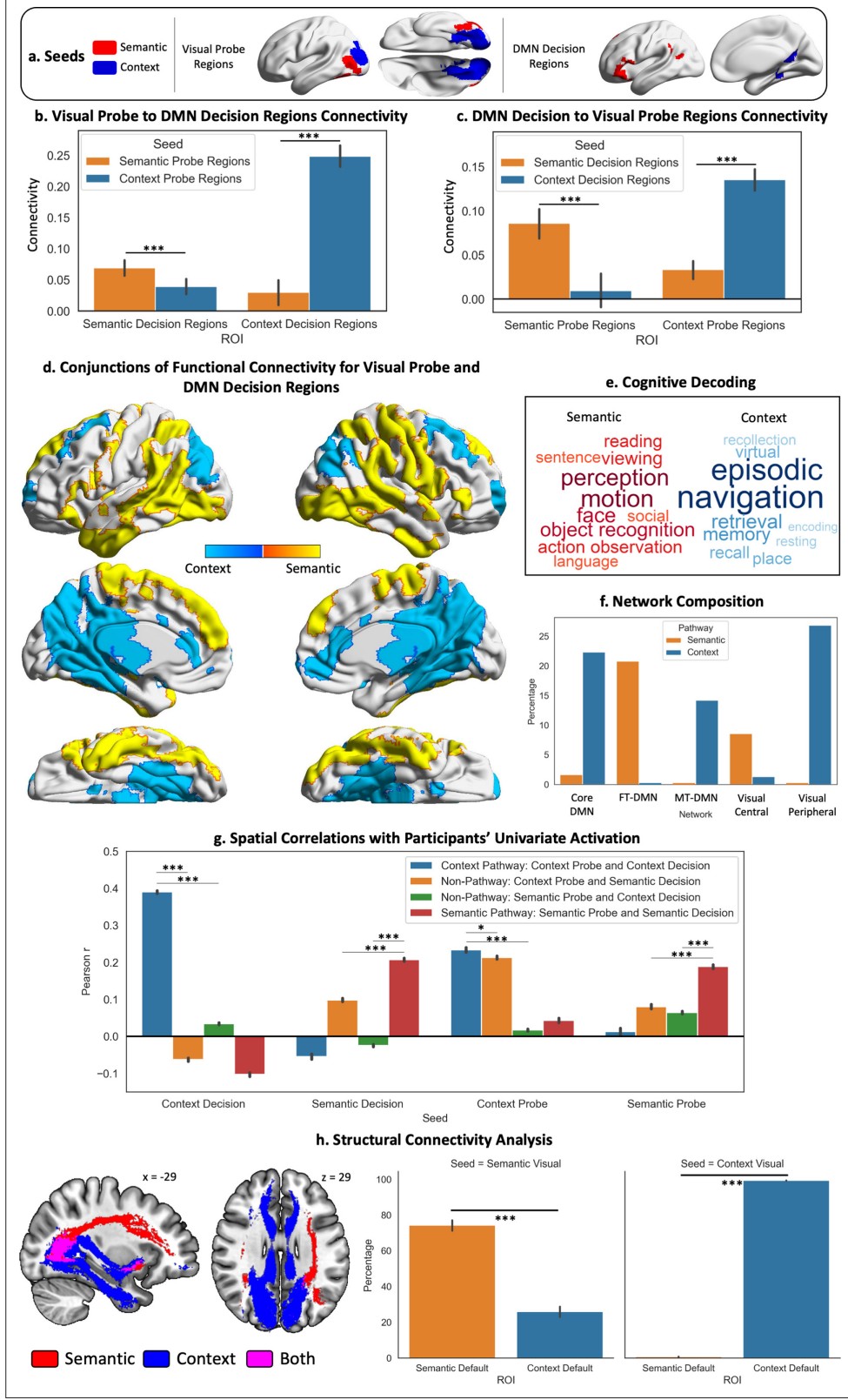

**Figure 4.** Visual to Default Network Pathways. (**a–c**) These panels depict the seeds, regions of interest (ROIs), and their connectivity. The bar plots in (**b and c**) show the connectivity between default mode network (DMN) decision regions and probe visual regions. (d) Warm colours = common regions showing stronger intrinsic connectivity to semantic decision regions in DMN and semantic probe regions in visual cortex; cool colours = common regions

*Figure 4 continued on next page*

*Figure 4 continued*

showing stronger intrinsic connectivity to spatial context decision regions in DMN and spatial context probe regions in visual cortex. (**e**) The cognitive decoding of these spatial maps using Neurosynth following the same colour code as (**d**). (**f**) Network composition showing the percentage of each pathway map overlapping with the three DMN and two visual subnetworks defined by the *Yeo et al., 2011*, 17-network parcellation. (**g**) Results of spatial correlation analysis comparing the semantic and spatial context pathways with non-pathway maps (derived from the conjunction of the connectivity of probe and decision seeds across different tasks, e.g., probe spatial context ∩ decision semantic connectivity). We assessed the spatial similarity of these pathway and non-pathway maps to the univariate activation during the probe and decision phases for each task and each participant. (**h**) Results of the structural connectivity analysis. Tracts displayed are a conjunction of streamlines between the probe and decision seeds of each task. The y axis of the bar plots shows the percentage of streamlines from each visual seed that terminate in each DMN ROI (shown in the x axis). The error bars depict the standard error of the mean. \*\*\*p<0.001, \*p<0.05.

The online version of this article includes the following figure supplement(s) for figure 4:

**Figure supplement 1.** Results of the re-analysis of intrinsic connectivity between semantic and context visual probe and default mode network (DMN) decision regions.

**Figure supplement 2.** Connectivity of the default mode network (DMN) decision regions to the object/scene localiser from Study 2, and that of the visual probe regions to fronto-temporal (FT), medial temporal (MT), and core DMN of Yeo's 17-network parcellation.

**Figure supplement 3.** The y axis of the bar plots shows the percentage of streamlines from each default mode network (DMN) seed that terminate in each visual region of interest (ROI) (shown in the x axis).

**Figure supplement 4.** Psychophysiological interaction analysis of the connectivity from the spatial context and semantic probe regions to fronto-temporal (FT) and medial temporal (MT)-default mode network (DMN) subnetworks.

cognition processing 'pathways'; (2) if this pattern was replicated in structural connectivity; (3) if it was present at the level of individual participants, and (4) we characterised the spatial layout, network composition (using influential RS networks), and cognitive decoding of these pathways. Having found dissociable pathways for semantic (object) and spatial context (scene) processing, we then examined their position in a high-dimensional connectivity space (*Figure 5*) that allowed us to document that the semantic pathway is less reliant on unimodal regions (i.e. more abstract) while the spatial context pathway is more allied to the visual system. Finally, we used uni- and multivariate approaches to examine how integration between these pathways takes place when semantic and spatial information is aligned (*Figure 6*).

## Probe phase

We began our exploration of the streams of information between visual cortex and DMN by characterising their visual end. To accomplish this, we first analysed the whole-brain activation observed during the probe phase of our semantic and spatial context tasks, when participants were viewing objects and scenes, and related these responses to previously established visual regions for object and scene perception. We then linked the visual regions engaged by our task to the DMN by describing their patterns of intrinsic connectivity, their functional involvement, and activation found within DMN regions during the probe phase of our task.

We examined differences in neural responses to probe images of objects in the semantic task, and scenes in the spatial context task in Study 1 (*Figure 2*). Semantic probes elicited greater activation in bilateral ventral LOC, extending to fusiform cortex and supramarginal gyrus in the left hemisphere. Spatial context probes elicited a stronger response in bilateral dorsal LOC, medial occipital lobe, precuneus, parahippocampal cortex and supplementary motor areas (SMA), as well as insula and middle frontal gyrus and frontal pole regions in the left hemisphere, and precentral regions in the right hemisphere (*Figure 2a*). Details of peak activations for all univariate results from Study 1 are in *Supplementary file 1* – Cluster information for neuroimaging results from Study 1.

To confirm these distinctive responses to semantic and spatial context probes were related to well-established categorical effects within visual cortex, we examined their overlap with object and scene localisers (i.e. passive viewing) in Study 2 (*Figure 2b and c*). Regions engaged by the spatial

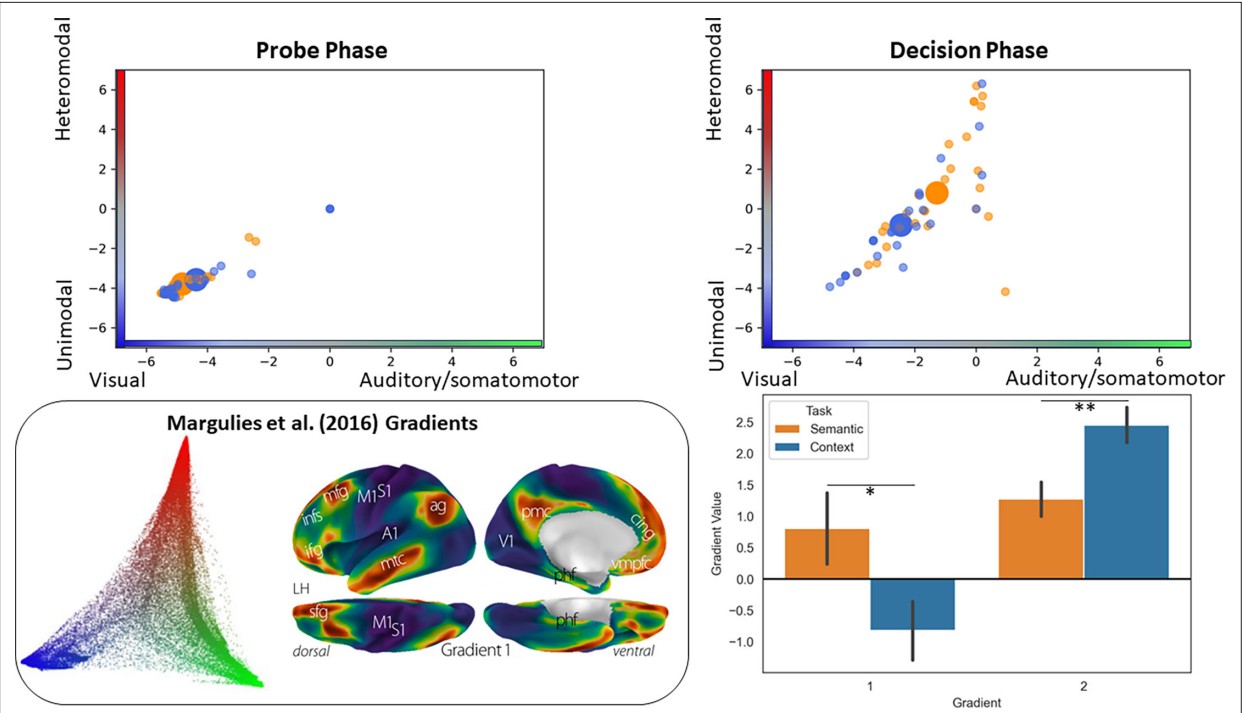

**Figure 5.** Analysis situating the position of the pathways in a whole-brain connectivity gradient space (*Margulies et al., 2016*). The scatterplots depict the position of each participant's peak response to the semantic and context task in this gradient space (the big circles represent the mean of each task for that phase). The bar plots compare the mean of each gradient across tasks. The inset on the bottom left of the panel displays *Margulies et al., 2016*, original gradient space.

The online version of this article includes the following figure supplement(s) for figure 5:

**Figure supplement 1.** Location in the two principal gradients of the peak response per participant for semantic and spatial context decisions.

context probes resembled scene perception regions, while semantic probes overlapped with object perception regions.

We next analysed the probe responses exploring the strength of activation during this phase within three DMN subdivisions defined by *Yeo et al., 2011*. [The Yeo et al. parcellation labels these subnetworks of the DMN as core DMN = DMN-A, FT-DMN=DMN-B, MT-DMN=DMN-C. A two-way repeated-measures ANOVA using task (semantic, spatial context) and region of interest (ROI) (core DMN, FT-DMN, MT-DMN) as factors revealed a significant main effect of ROI (F(1.281,33.306)=50.42, p<0.001) and an interaction (F(1.64,42.65)=12.44, p<0.001; Greenhouse-Geisser corrected).] Post hoc comparisons showed a significantly stronger response within MT-DMN to spatial context relative to semantic probes (t(26)=4.1, p=0.001). No significant difference between tasks was observed for core or FT-DMN (both p>0.05, *Figure 2d*).

Finally, we examined the intrinsic connectivity of activation regions in *Figure 2a*, masked by *Yeo et al., 2011*, visual networks (combining central and peripheral networks), using data from Study 3. Visual areas responding to semantic and spatial context probes showed differential connectivity, including to regions of DMN (posterior cingulate, medial prefrontal cortex, portions of anterior and dorsal prefrontal cortex, and anterior temporal cortex; *Figure 2e*). Cognitive decoding using Neurosynth revealed that semantic probe connectivity was associated with perceptual and somato-motor terms, while spatial context probe connectivity was associated with navigation, visuospatial and episodic memory terms (*Figure 2f*). Semantic probe regions showed preferential overlap with FT-DMN, whilst spatial context probe regions showed greater overlap with MT-DMN, followed by core DMN (*Figure 2g*).

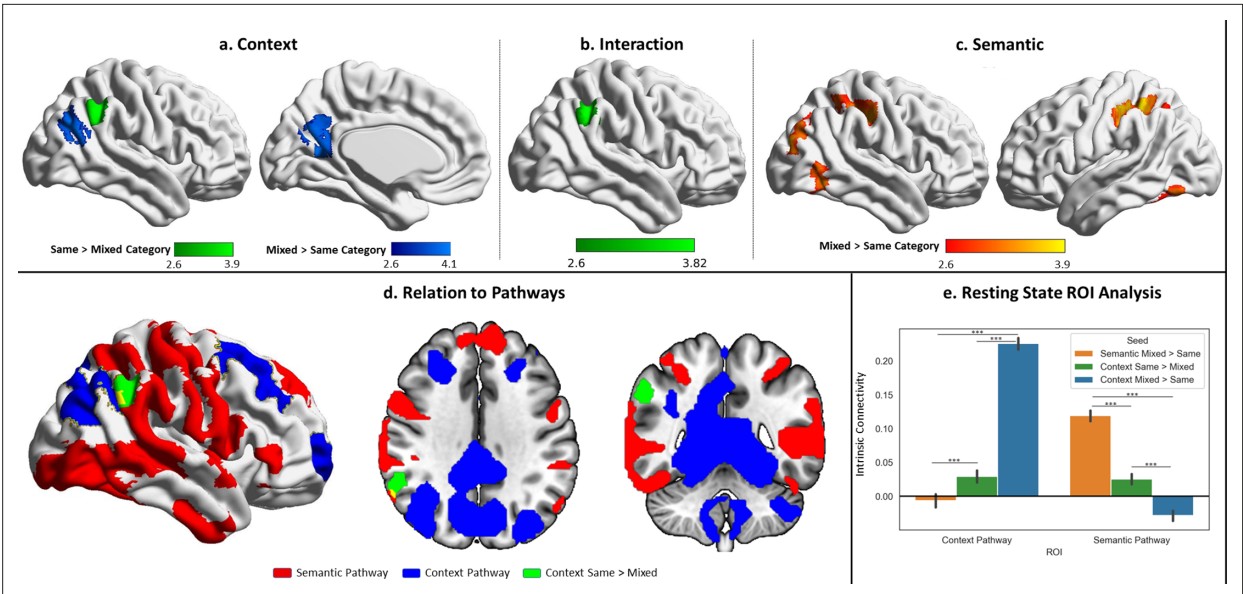

**Figure 6.** Univariate results for the probe phase contrasting same-category versus mixed-category trials separately for the semantic and spatial context tasks. (**a**) Contrast of spatial context same- > mixed-category building trials during the probe phase. (**b**) Task by condition (same/mixed-category building) interaction. (**c**) Contrast of semantic same- > mixed-category building trials during the probe phase. (**d**) Spatial relations of the same > mixed spatial context cluster with the semantic and spatial context pathways outlined in *Figure 4*. (**e**) Intrinsic connectivity seed-to-region of interest (ROI) results using the three univariate results clusters shown in the top panel as seeds and the pathways as ROIs. The error bars depict the standard error of the mean. ***p<0.001.

The online version of this article includes the following figure supplement(s) for figure 6:

**Figure supplement 1.** Results of the representational similarity analysis (RSA).

**Figure supplement 2.** Representational similarity analysis results for the same-category building trials of the probe phase of the semantic task.

## Decision phase

Having characterised the visual end of the visual-DMN pathways in our previous analysis, we next turned our attention to the DMN end. Following the same logic, we first analysed the whole-brain activation during the decision phase of our task, where participants had to judge the relationship between objects and scenes respectively. We compared this activation to classic DMN regions described by *Yeo et al., 2011*, and linked the regions engaged within this network to visual regions in terms of their task activation and functional connectivity.

We characterised regions responsive to semantic and spatial context decisions in Study 1. *Figure 3a* shows that semantic decisions elicited stronger engagement within dorsolateral prefrontal, lateral occipital, posterior temporal, and occipital cortex, as well as pre-SMA. These regions overlapped with FT-DMN (*Figure 3b*). Spatial context decisions produced stronger activation within a predominantly medial set of occipital, ventromedial temporal (including parahippocampal gyrus), retrosplenial, and precuneus regions that overlapped with MT-DMN (*Figure 3c*).

Given our hypothesis of dissociable pathways between visual cortex and DMN subsystems, we examined responses to semantic and spatial contextual decisions in visual cortex, using the object and scene localiser ROIs from Study 2. A two-way repeated-measures ANOVA including task (semantic, spatial context decisions) and visual ROI (scene and object regions) revealed main effects of task (F(1,26)=14.02, p<0.001), visual ROI (F(1,26)=9.91, p=0.004), and their interaction (F(1,26)=24.65, p<0.001). Post hoc comparisons showed a stronger response to spatial context decisions, relative to semantic decisions, in the visual ROI sensitive to scenes (t(26)=3.63, p<0.001), but not in the object-selective region (t(26)=1.88 p=0.071; *Figure 3d*).

We investigated differences in the intrinsic connectivity of the distinct DMN decision regions activated during the semantic and spatial context tasks (shown in *Figure 3a*) in independent data from Study 3. We seeded the decision regions that intersected with DMN (combining core, FT and MT-DMN from Yeo et al.'s 2011 parcellation). The results, in *Figure 3e*, revealed differences in the functional

networks of these DMN regions that extended to visual cortex. Semantic decision regions showed stronger connectivity to lateral visual regions along with lateral temporal cortex, inferior frontal gyrus, angular gyrus, and dorsomedial prefrontal cortex. Spatial decision regions were more connected to medial visual regions, ventro-medial temporal regions, medial parietal cortex, ventral parts of medial prefrontal cortex, motor cortex and dorsal parts of LOC. Since resting-state analysis is sensitive to the choice of threshold, we repeated this analysis with a stricter cluster-forming threshold and found that the resulting maps were virtually identical (with a spatial correlation of r=0.99 between maps generated with both thresholds), for both the probe and decision phases (see Supplementary analysis: resting-state maps with stricter thresholding).

Cognitive decoding of these connectivity maps using Neurosynth (*Figure 3f*) revealed that the semantic decision network was associated with terms related to language, semantic processing, and reading, while the spatial context decision network was associated with navigation, episodic, and autobiographical memory. The decision DMN seeds also showed differential connectivity to visual regions (*Figure 3g*). Semantic decision regions were more connected with lateral and ventral occipital cortex, whilst spatial context decision regions showed more connectivity with medial occipital, ventro-medial temporal (including parahippocampal), and dorsal LOC.

## Pathways analysis

The analysis above identified regions of visual cortex showing a differential response to semantic and spatial context probes, related to category effects for objects versus scenes. We also found distinct DMN subnetworks which supported semantic and spatial context decisions respectively. Next, we considered if these effects are linked: Do FT-DMN regions have stronger connectivity to object perception areas of visual cortex, while MT-DMN regions connect to scene perception regions? To answer this question, we analysed the functional and structural connectivity from the visual regions that were responsive to viewing probes during our tasks to those DMN regions activated during decisions about those probes. We characterised the spatial layout of these pathways in the brain as well as their large-scale network composition, their functional involvement, and verified their presence in individual participants.

Using resting-state data from Study 3, we performed a series of seed-to-ROI analyses to examine differential visual-to-DMN connectivity. We seeded the visual regions in *Figure 2e* (i.e. probe responses to objects and scenes masked by Yeo et al.'s visual networks) and extracted their intrinsic connectivity to DMN, using ROIs showing differential activation to semantic and spatial context decisions (corresponding to the seeds in *Figure 3e*). In a second analysis, we examined the reverse (i.e. seeded DMN regions and extracted their connectivity to visual regions). The seeds and ROIs for this analysis can be consulted in *Figure 4a*. These effects were analysed using repeated-measures ANOVAs examining the interaction between seed and ROI (*Figure 4b and c*). The visual-to-DMN ANOVA showed main effects of seed (F(1,190)=226.23, p<0.001), ROI (F(1,190)=85.21, p<0.001), and a seed by ROI interaction (F(1,190)=322.83, p<0.001). Post hoc contrasts confirmed there was stronger connectivity between object probe regions and semantic versus spatial context decision regions (t(190)=3.98, p<0.001), and between scene probe regions and spatial context versus semantic decision regions (t(190)=20.07, p<0.001). The DMN-to-visual ANOVA confirmed this pattern: again, there was a main effect of ROI (F(1,190)=36.91, p<0.001) and a seed by ROI interaction (F(1,190)=218.42, p<0.001), with post hoc contrasts confirming stronger intrinsic connectivity between DMN regions implicated in semantic decisions and object probe regions (t(190)=11.63, p<0.001), and between DMN regions engaged by spatial context decisions and scene probe regions (t(190)=6.17, p<0.001). To ensure that these results were not artificially inflated due to spatial mixing of the resting-state signals arising from proximal visual peripheral and DMN-C networks (*Silson et al., 2019*; *Steel et al., 2021*), we conducted a supplementary analysis eroding the visual probe and DMN decision ROIs for the spatial context task until the minimum gap between them exceeded the size of our smoothing kernel (see Supplementary analysis: eroded masks replication analysis and *Figure 4—figure supplement 1*). The results replicated the pattern described above. Supplementary analyses using the same seeds and task-independent ROIs also revealed the same pattern: these ROIs were based on the visual localiser masks from Study 2 and the complete DMN subnetworks defined by the *Yeo et al., 2011*, 17-network parcellation (see Supplementary analysis: replicating resting-state connectivity pathways with task-independent ROIs and *Figure 4—figure supplement 2*).

These pathways, specialised for semantic and spatial cognition, link dissociable visual regions to DMN subsystems, consistent with the suggestion that functional differentiation in DMN partly reflects the strength of different inputs. *Figure 4d* provides a visualisation of these pathways using the intersection of connectivity from object over scene probe regions and semantic versus spatial context decisions to identify the semantic pathway (warm colours) and the reverse contrasts for the spatial pathway (cool colours). Cognitive decoding revealed terms related to object, action, motion, social, and face perception, as well as language and reading for the semantic pathway, and terms related to navigation, place processing and memory for the spatial context pathway (*Figure 4e*). The semantic pathway was predominantly characterised by FT-DMN and visual central regions in the *Yeo et al., 2011*, 17-network parcellation, whilst the spatial context pathway reflected core and MT-DMN, and visual peripheral networks (*Figure 4f*).

A complementary analysis examined the spatial correlation of these semantic and spatial pathways (*Figure 4d*) with participants' univariate activation when viewing semantic and spatial context probes, and when making semantic and spatial context decisions. We compared spatial correlations between our hypothesised pathways and 'non-pathway conjunctions', defined as conjunctions of visual object probe and DMN spatial context decision connectivity, and visual scene probe and DMN semantic decision connectivity with these univariate activation maps. We obtained Pearson r values for each participant reflecting spatial similarity of their activation patterns with these pathway and non-pathway maps and compared these correlations using one-way ANOVAs (*Figure 4g*). There was a significant effect of pathway for each of the four task phases (spatial context decision: $F(2.32,441.45)=2741.68$, $p<0.001$; semantic decision: $F(1.90,361.29)=521.94$, $p<0.001$; spatial context probe: $F(2.09,396.96)=424.98$, $p<0.001$; semantic probe: $F(2.07,393.80)=117.88$, $p<0.001$; Greenhouse-Geisser correction applied). In follow-up contrasts using paired t-tests, we compared the average Pearson r correlation of each phase to its relevant pathway, contrasted with the two non-pathway conjunctions. Correlations between the semantic probe and decision phases' univariate activation and the semantic pathway were higher than non-pathway correlations and an equivalent pattern was seen for the spatial context pathway (*Figure 4g*; for exact t and p values associated with these comparisons see *Supplementary file 2*– Paired t-tests contrasting spatial similarity of participant-level activation with group-level context and semantic pathways and non-pathways), confirming that dissociable visual-to-DMN responses associated with semantic and spatial cognition are reliably present for individual participants.

Next, we examined if these pathways were reflected in the strength of white matter tracts connecting visual and DMN regions (seeds in *Figure 4a*). We examined structural connectivity in a subset of the Human Connectome Project (HCP) dataset (n=164), asking if object probe visual regions showed a greater proportion of white matter tracts terminating in semantic DMN regions, and if scene probe visual regions showed stronger structural connectivity to spatial context DMN regions (*Figure 4h*). A 2×2 repeated-measures ANOVA with visual regions as seeds and DMN regions as ROIs revealed a significant main effect of seed ($F(1,163)=5.13$, $p=0.025$), ROI ($F(1,163)=82.46$, $p<0.001$), and their interaction ($F(1,163)=664.57$, $p<0.001$). Post hoc comparisons confirmed stronger structural connectivity from the semantic probe visual regions to the semantic decision DMN regions; likewise, the spatial context probe visual regions showed stronger structural connectivity to the spatial context decision DMN regions (semantic probe visual: $t(163)=8.25$, $p<0.001$; context probe visual: $t(163)=478.66$, $p<0.001$). Repeating this analysis using the decision DMN regions as seeds and the probe visual regions as ROIs revealed a similar pattern (see Supplementary analysis: replicating pathways' structural connectivity from the DMN end and *Figure 4—figure supplement 3*).

Finally, we examined how connectivity within the pathways changes depending on task demands in a psychophysiological interaction (PPI) analysis (see supplementary materials). We took the visual regions showing differential activation to object and scene probes as seeds (shown in *Figures 2e and 4a*), while the ROIs were regions sensitive to semantic and spatial context decisions within the DMN (shown in *Figures 3e and 4a*). The results, shown in *Figure 4—figure supplement 4*, showed that the object seed was more connected to both semantic and spatial context DMN decision regions during the semantic task, while the scene probe regions were more connected to spatial context decision regions during the spatial context task than object probe regions.

## Location of pathways in whole-brain gradients

Having found evidence for dissociable semantic and spatial context pathways, we analysed their location in a functional state space defined by the first two gradients of intrinsic connectivity (*Margulies et al., 2016*). The principal gradient relates to connectivity differences between unimodal and heteromodal cortex, while the second gradient captures connectivity differences between visual and auditory/somatomotor cortex. By locating the ends of the two visual-to-DMN pathways within gradient space, we can establish if DMN regions supporting semantic and spatial cognition are equally distant in connectivity from sensory-motor cortex: semantic cognition is arguably more abstract than spatial cognition and might be supported by DMN regions that are more isolated from sensory-motor systems on the principal gradient (*Margulies et al., 2016*; *Smallwood et al., 2021*). We can also establish if semantic and spatial DMN regions differ in the balance of connectivity to visual versus auditory-motor regions on the second gradient: heteromodal concepts are thought to be constructed from diverse sensory-motor features (*Ralph et al., 2017*), while spatial representations might draw more strongly on visual information (*Epstein and Baker, 2019*). We tested these predictions by locating individual unthresholded peak response coordinates for semantic and spatial context probes (within visual networks) and decisions (within the DMN) in gradient space (masked by Yeo et al.'s 7-network parcellation). We then asked if there are significant differences in the gradient locations of these tasks across participants.

The results (*Figure 5a*) showed that there were no differences between the two tasks during the probe phase, while the decision phase was associated with task effects: DMN peaks for semantic decisions were more distant from sensory-motor cortex on the principal gradient, compared with spatial context decisions (t(1,26)=2.34, p=0.027), consistent with the view that semantic cognition draws on more abstract and heteromodal representations in DMN. In addition, responses for the spatial context task were closer to the visual end of the second gradient, while responses for the semantic task were somewhat more balanced across visual and auditory-motor ends of this gradient (t(1,26)=3.31, p=0.003). [An ANOVA including task and condition (MCB versus SCB) replicated these task effects and found no effects of condition on the position of peak responses in gradient space.] Since the scatterplots in *Figure 5* do not distinguish whether these effects took place at the individual level (the data points are not linked across tasks), we plotted the same data comparing the gradient values for the peak responses in each of our tasks at the participant level in the supplementary materials (see *Figure 5—figure supplement 1*). This plot shows that in the majority of individual cases, the pattern of group-level results shown in *Figure 5* held.

## Cross-pathway integration of semantic and spatial cognition: response to SCB versus MCB

Having identified dissociable semantic and spatial context pathways, and examined how these are differentially recruited across tasks, we investigated the integration of semantic and spatial context information across these processing streams. We compared responses in SCB and MCB trials, since semantic and spatial information are aligned when buildings contain items from a single semantic category, but not in MCB. In these analyses, there were differences between conditions in the probe but not the decision-making phase (perhaps because many probes were presented without decisions, increasing statistical power).

First, we performed univariate contrasts of MCB and SCB trials (*Figure 6a–c*). For scene probes in the spatial context task, the MCB>SCB contrast elicited a stronger response in dorsal LOC and retrosplenial cortex (*Figure 6a*). In these circumstances, spatial context probes could only activate spatial and not semantic information. The SCB>MCB contrast activated an adjacent region of right angular gyrus (*Figure 6b*). For semantic probes, the contrast of MCB>SCB identified greater engagement in distributed parietal, occipital, and temporal regions, associated with the multiple-demand network (*Figure 6c*). There were no clusters that showed a stronger response to SCB than MCB probes for the semantic task. There was also an interaction between task and condition, which was driven by the SCB>MCB effect in right angular gyrus in the spatial context task exceeding this effect in the semantic task.

To interpret these results, we conducted seed-to-ROI intrinsic connectivity analysis using independent data from Study 3, taking these clusters as seeds and the semantic and spatial context pathways masks (shown in *Figures 4d and 6d*) as ROIs. A two-way repeated-measures ANOVA

examined seed (spatial context SCB>MCB and MCB>SCB; semantic MCB>SCB) and ROI (semantic and spatial context pathways) as factors. The results can be seen in *Figure 6e*. There were significant effects of seed (F(1.92,364.1)=211.48, p<0.001), ROI (F(1,190)=182.67, p<0.001) and an interaction (F(1.84,349.66)=723.412, p<0.001). Post hoc comparisons showed the context pathway was most connected to the spatial context MCB>SCB clusters, less connected to the spatial context SCB>MCB (when semantic information was also relevant to the response), and least connected to the semantic MCB>SCB regions (context MCB>context SCB: t(1,190)=34.91, p<0.001; context SCB>semantic MCB: t(1,190)=5.18, p<0.001). The opposite pattern of connectivity was found for the semantic pathway, which was most connected to the semantic MCB>SCB regions, less connected to the spatial context SCB>MCB, and least connected to the spatial context MCB>SCB clusters (semantic MCB>context SCB: t(1,190)=16.93, p<0.001; context SCB>context MCB: t(1,190)=10.59, p<0.001). In this way, spatial context SCB>MCB regions, which reflected the engagement of semantic information in a spatial context task, showed an intermediate pattern of connectivity to both pathways.

A supplementary analysis of the task using multivariate approaches (see supplementary materials 'Supplementary analysis: multivariate response to SCB versus MCB') found a similar pattern. We performed representational similarity analysis (RSA) using a searchlight approach, which allowed us to detect regions sensitive to semantic and spatial context information. The results of this analysis in the MCB trials, where information could not be integrated, showed regions sensitive to category in the semantic task in bilateral ventral LOC, as well as regions sensitive to location in the spatial context task, dorsally in left LOC (see *Figure 6—figure supplement 1*). A cross-task RSA of the SCB trials, where semantic and spatial information were aligned, allowing integration, identified a separate set of regions in right LOC, topographically situated between the two pathways, that captured spatial context information during the semantic task. An intrinsic connectivity analysis in data from Study 3 using the RSA regions as seeds and the pathways as ROIs showed that the regions sensitive to semantic and spatial context information during MCB trials were maximally connected to the semantic and spatial context pathways, respectively. On the other hand, the regions where spatial information could be decoded during the SCB trials of the semantic task showed an intermediate pattern of connectivity to both pathways (especially to the semantic pathway), suggesting a role in integrating information between them (see *Figure 6—figure supplement 1*).

## Discussion

Functional subdivisions of visual cortex and DMN sit at opposing ends of parallel processing streams supporting visually mediated object-centric semantic and spatial cognition; moreover, regions with intermediate patterns of connectivity are implicated in the integration of these streams into coherent experience. Viewing object probes in a semantic task and location probes in a spatial context task activated different parts of visual cortex, functionally related to the passive viewing of objects and scenes. Semantic and spatial context decisions about these probes engaged FT and MT-DMN subsystems. Visual regions sensitive to object probes showed stronger intrinsic functional connectivity and structural connectivity to FT-DMN, while scene probe regions were more connected to MT-DMN. In a functional space defined by whole-brain connectivity patterns, the object-centric semantic pathway was more distant from unimodal regions on a unimodal-to-heteromodal connectivity gradient, and it had a more balanced influence of visual and auditory-motor systems, while the spatial context pathway was more visual. Finally, we found evidence that both heteromodal and visual regions integrate information about meaning and spatial context. When all the items in a building were drawn from a particular semantic category, there was greater recruitment of right angular gyrus; multivariate pattern analysis similarly found a cluster in LOC that represented spatial context information during the semantic task. When there was no opportunity to integrate object-centric semantic information with spatial context, regions that responded showed higher pathway-specific connectivity. In contrast, when integration was facilitated by the structure of the task, response regions had an intermediate pattern of connectivity.

Our study has important implications for the organisation of DMN into specialised subsystems, and for how these subsystems get their input from perceptual regions. Previous literature has robustly established distinct FT and MT subsystems (*Andrews-Hanna et al., 2014*; *Andrews-Hanna and Grilli, 2021*; *Smallwood et al., 2021*; *Yeo et al., 2011*); however, the way in which this architecture reflects differences in visual inputs remains contentious. One proposal is that different DMN subnetworks are

differently engaged by tasks that are externally versus internally oriented. For example, (*Chiou et al., 2020*) propose that there is a basic distinction between parts of the network that process semantic information accessed from words and images, and between DMN regions that sustain internally focussed cognition. Other work has called into question whether semantic responses in FT-DMN are specific to external tasks: e.g., *Zhang et al., 2022*, found that lateral temporal regions changed their patterns of connectivity depending on the task, with more visual connectivity in externally oriented tasks like reading, and more DMN connectivity in internally orientated conceptual states like mind-wandering and autobiographical memory. This work suggests FT-DMN might support object-centric semantic cognition across internal and external modes of cognition. Our findings also suggest that the distinction between these subsystems is not organised according to visual coupling; instead, DMN organisation arises from differential connectivity between distinct visual and DMN regions that gives rise to partially segregated pathways that process information about locations and meanings. Visual responses to scenes and objects reflect entry points to these processing pathways such that the key distinction between FT-DMN and MT-DMN relates to the type of information being processed, as opposed to how the information is accessed.

Our observed dissociation between semantic and spatial context pathways echoes a similar domain-specific organisation for working memory in prefrontal cortex (*Levy and Goldman-Rakic, 2000*; *Romanski, 2004*), in which there are dorsal and ventral streams associated with the maintenance of item location and identity respectively. This organising principle has been extended to long-term memory more recently. *Deen and Freiwald, 2021*, found a similar dissociation between places and people (instead of objects) in the DMN and other areas of association cortex. This was not tied to a specific input modality or task, indicative of parallel, domain-specific networks, at the top of the cortical hierarchy. Here, we extend this approach to consider whether functional divisions within DMN and visual cortex are connected, giving rise to pathways which are differentially situated in a connectivity state space defined by whole-brain dimensions of intrinsic connectivity, and we ask how these pathways might be integrated, and how they might be flexibly recruited according to task demands.

Although our research suggests a *domain*-specific view of brain organisation within visual-to-DMN pathways linked to object-centric semantic and spatial cognition, there may be different *processes* within meaning and spatial context tasks that drive these effects. The FT subsystem is thought to rely on the abstraction of information from sensory-motor inputs (; *Smallwood et al., 2021*; *Wang et al., 2020*). The MT subsystem, on the other hand, uses a relational code that can capture spatial relations to successfully navigate complex environments (*Eichenbaum, 2004*; *Eichenbaum and Cohen, 2014*; *Zeidman et al., 2015*). One possibility is that, at the visual end of these pathways, spatial location is more dependent on peripheral vision, while object recognition is dependent on central fixation (*Hasson et al., 2002*; *Levy et al., 2001*); consequently, the distinct visual-to-DMN pathways we have recovered may reflect a basic property of how peripheral and central visual regions project to DMN. Our findings mirror and extend the results of *Silson et al., 2019*; *Steel et al., 2021*, since we identify dissociable pathways between visual cortex and DMN; however, we extend this work to cover fully distributed networks that support object-centric semantic and spatial decision-making, and locate these pathways in a whole-brain gradient space relating to variation in patterns of intrinsic connectivity, as well as considering how these pathways can be integrated.

One question remains: how does the brain generate a coherent, seamlessly integrated experience of place and the identity of objects from these segregated, specialised streams of processing? The response we identified in right angular gyrus when object semantic and spatial context information was aligned is consistent with earlier studies implicating this brain region in the integration of information from multiple domains into a rich, meaningful context that can guide ongoing cognition (*Lanzoni et al., 2020*). One recent proposal suggests that neurons in this area represent high-dimensional inputs on a low-dimensional manifold encoding the relative position of items in physical space and abstract conceptual space (*Summerfield et al., 2020*). This region, which is maximally distant from sensory-motor cortex and equidistant from visual and motor cortex, might have the capacity to form representations that are not dominated by one type of input or code. We found a region of the right AG that was potentially important for integrating semantic and spatial context information. Previous research has established a key role of the AG in context integration (*Bonnici et al., 2016*; *Branzi et al., 2020*; *Ramanan et al., 2018*) and specifically, in guiding multimodal decisions and behaviour (*Humphreys et al., 2021*; *Xu et al., 2017*; *Yazar et al., 2017*). Although some recent proposals

suggest a causal role of right AG in the early establishment of meaningful contexts, allowing semantic integration across modalities (*Bocca et al., 2015*; *Muggleton et al., 2008*; *Olk et al., 2015*; *Petitet et al., 2015*; *Seghier, 2023*), the majority of this research points to left, rather than right, AG as a key region for integration. We might have observed involvement of the right AG in our study since people were integrating semantic and spatial information, and visuospatial memory processes might be somewhat right lateralised (cf. *Sormaz et al., 2017*) and more strongly connected to right than left AG. We are not aware of a literature on right AG lesions impairing the integration of semantic and spatial information but, in the face of our findings, this might be a promising new direction. Patients with damage to right AG should be examined with specific tasks aimed at probing this type of integration. We also found evidence of information integration in occipital regions that were closer to the input regions of the visual-to-DMN pathways. These different levels of integration shared a common characteristic: in both cases, the region implicated in integration was spatially interposed between the pathways, consistent with the view that topography is highly relevant to information integration since adjacent brain regions tend to share a high degree of functional connectivity and represent similar information.

While we might assume that common visual-to-DMN pathways support memory access from vision (as in this study), and subserve the generation of visual features when imagining objects versus scenes, this hypothesis awaits empirical investigation. Moreover, our pathways are vision-specific, and it remains unclear if there are analogous pathways from auditory or somatomotor cortex to DMN. The generality of these pathways must be confirmed across tasks, since spatial representations are likely to interact with other representational codes, including emotion and social information – the interdigitated pathways highlighted in these circumstances (*Braga and Buckner, 2017*; *DiNicola et al., 2020*) might show anatomical differences or be broadly the same as the pathways uncovered here.

Likewise, further research should be carried out on memory-visual interactions for alternative domains. Our study focussed on spatial location and semantic object processing and therefore cannot address how other categories of stimuli, such as faces, are processed by the visual-to-memory pathways that we have identified. Previous work has suggested some overlap in the neurobiological mechanisms for semantic and social processing (; *Chiou et al., 2020*), suggesting that the FT-DMN pathway may be highlighted when contrasting both social faces and semantic objects with spatial scenes. On the other hand, some researchers have argued for a 'third pathway' for aspects of social visual cognition (*Pitcher, 2023*; *Pitcher and Ungerleider, 2021*). Future studies that probe other categories will be able to confirm the generality (or specificity) of the pathways we described.

One important caveat is that we have not investigated the spatiotemporal dynamics of neural propagation along the pathways we identified between visual cortex and DMN. The dissociations we found in task responses, intrinsic functional connectivity, and white matter connections all support the view that there are at least two distinct routes between visual and heteromodal DMN regions, yet this does not necessarily imply that there is a continuous sequence of cortical areas that extend from visual cortex to DMN – and given our findings of structural connectivity differences that relate to the functional subdivisions we observe, this is unlikely to be the sole mechanism underpinning our findings. It would be interesting in future work to characterise the spatiotemporal dynamics of neural propagation along visual-DMN pathways using methods optimised for studying the dynamics of information transmission, like Granger causality or travelling wave analysis.

Moreover, many questions remain about information integration across the semantic and spatial domains; does spatial juxtaposition promote the emergence of an integrated code, and are neural representations that emerge at the intersection of these pathways more than the sum of their parts? Although further research is needed, the current study highlights how subdivisions within visual and DMN are related to types of information, giving rise to distinct processing streams that capture different unimodal to heteromodal transformations relevant to object-centric semantic and spatial context processing, and shows how these pathways might interact at multiple levels of the cortical hierarchy to produce coherent cognition.

## Methods
### Study 1: Task-based fMRI
### Study 1: Participants

Thirty native English speakers (mean age = 22.6 ± 2.7 years, age-range 18–34 years, 8 males) with normal or corrected-to-normal vision and no history of language disorders participated in this study. Ethical approval was obtained from the Research Ethics Committees of the Department of Psychology and York Neuroimaging Centre, University of York. Written informed consent was obtained from all subjects prior to testing.

### Study 1: Materials

The learning phase employed videos showing a walk-through for 12 different buildings (one per video), shot from a first-person perspective. The videos and buildings were created using an interior design program (Sweet Home 3D). Each building consisted of two rooms: a bedroom and a living room/office, with an ajar door connecting the two rooms. The order of the rooms (first and second) was counterbalanced across participants. Each room was distinctive, with different wallpaper/wall colour and furniture arrangements. The building contexts created by these rooms were arbitrary, containing furniture that did not reflect usual room distributions (i.e. a kitchen next to a dining room), to avoid engaging further conceptual knowledge about frequently encountered spatial contexts in the real world. Within each room, there were three framed images of objects and animals, towards the start, middle, and end of each video (see top panel of *Figure 7—figure supplement 1*), each at the same distance from its neighbour (across the two rooms); 72 images were presented overall. The images represented single items from six semantic categories: musical instruments, gardening tools, sports equipment, mammals, fish, and birds (12 pictures from each category). Half of the buildings contained images from the same semantic category (SCB), and the other half contained images from different semantic categories ('MCB). The presentation of items within each room in MCB was controlled such that: (1) no item from the same category was presented at the same location twice, and (2) no three categories were grouped together more than once.

A full list of pictures of the object and location stimuli employed in this task as well as the videos watched by the participants can be consulted in the OSF collection associated with this project under the components OSF > Tasks > Training.

### Study 1: Design and procedure
#### Training task

Subjects participated in a training session the day before the MRI scan, where they watched the walk-through videos, each lasting 49 s (*Figure 7* and top panel of *Figure 7—figure supplement 1*). Participants watched each video at least six times in three rounds (twice per round). Each round consisted of four mini-blocks of videos containing three videos. After each mini-block, participants were given a test in Psychopy3: they were asked to choose the room that each item was presented in, responding via button press. Items were pictured at the top of the screen, with the correct room and another room below (*Figure 7—figure supplement 1*, left half of bottom panel). Screenshots were taken of the location of each framed image and the rooms themselves (from the entrance way), with the images and their frames removed (see bottom panel of *Figure 7—figure supplement 1*). They had 5 s to respond, after which the correct room was presented as feedback for a further 5 s. Following each round, there was a matching task, which reinforced participants' memory of which rooms belonged together. Two rooms from the same building were presented, with two items from that building (one from each room) below. Participants were instructed to drag the objects into the correct room of the building (*Figure 7—figure supplement 1*, right half of bottom panel). Feedback showed the correct object in each room. Finally, to establish how well the item-location pairs were learned, participants were given a final test on all the rooms and items: this followed the structure of the mini-block tests, except that materials from the entire session were included. If accuracy was below 80%, participants watched the videos again until this threshold was reached. In total, participants spent approximately 2 hr on the training. The amount of training required was established in pilot testing with nine participants who did not take part in the main study. This also confirmed the items were easily nameable.

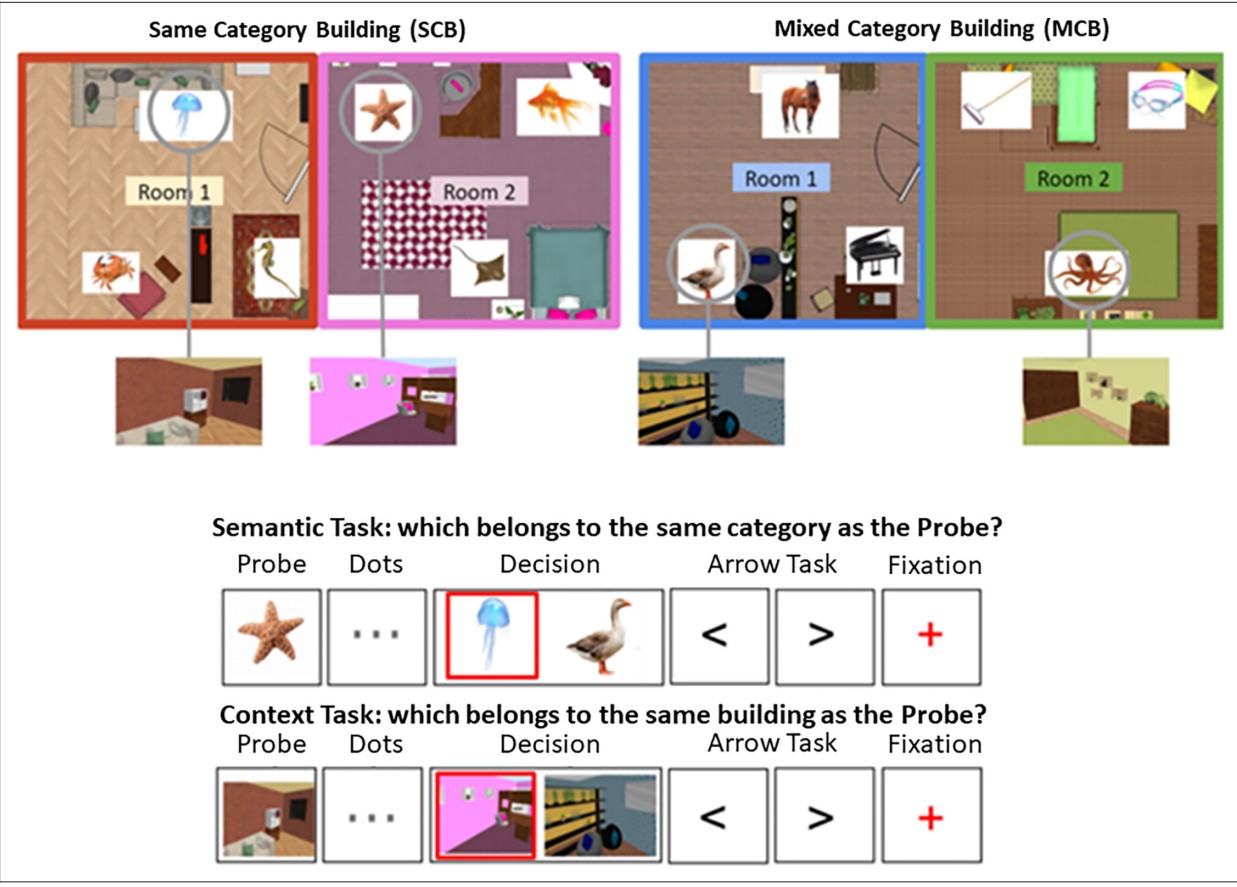

**Figure 7.** Top panel: the layout of two buildings is shown, one of them contains semantically related items (same-category building [SCB]), the other contains unrelated items (mixed-category building [MCB]). These items and locations are shown in the example trials below. Bottom panel: Trial procedure for semantic and spatial context decisions. The phases of a trial are shown (probe, dots, decision, arrow task, fixation), and the red square indicates the correct response (not shown to participants). Participants were required to press 'left' or 'right' buttons in the decision phase. No-decision trials omitted the dots and decision phases. The videos and tests used during the training session for this task as well as the stimuli used in the task of Study 2 can be consulted in *Figure 7—figure supplements 1 and 2*.

The online version of this article includes the following figure supplement(s) for figure 7:

**Figure supplement 1.** Top panel: An example of what was seen during the training videos.

**Figure supplement 2.** Examples of the dynamic images taken from the 3 s movie clips depicting faces, bodies, scenes, objects, and scrambled objects.

## fMRI task

On the day of the scan, participants repeated the final test from the training day to establish how well they retained the information. They then watched all 12 videos once again, in a counterbalanced order, and performed the test phase a second time. The mean accuracy on the first day was 95.1% (SD = 5.7%) while on the second day it was 97.1% (SD = 3.8%). Two participants were excluded from the mean accuracy calculation for day 2 due to data loss.

Inside the scanner, participants performed a semantic and spatial context memory task, using a slow event-related design (*Figure 7*). The semantic task involved judgements about the semantic category of objects and animals (using the same images that have been presented in the buildings), while the spatial context task involved matching rooms that belonged to the same building. Both tasks consisted of 'no-decision' and 'decision' trials. No-decision trials were optimised for RSA and included an image of the probe item (for 2 s) followed by an 'arrow task' in which participants pressed 'left' or 'right' to match the direction of a series of chevrons ('<' '>') presented on the screen (for 3 s), ending with a red fixation cross (1 s). The probe in the semantic task was an object or animal from the training; for the spatial context task, it was a screenshot of an item's location within a room (excluding the item itself). In decision trials, the same types of probes were presented but

they were followed by three central dots (for 4 s) indicating a decision would be made (**Figure 7**). In the decision phase, a target image (from the same category or building as the probe) was presented together with a distractor image, creating a two-alternative forced-choice judgement. In the spatial context task, the distractor was a room from a different building, while in the semantic task, the distractor was an item from a different category. On SCB trials, semantic and spatial information were aligned in the sense that the target and probe were from the same category and the same building. On MCB trials, semantic and spatial information did not converge: semantic targets were in a different building from the probe, while spatial context targets were from a different semantic category from the probe (see **Figure 7** for the structure of semantic and spatial context trials). Decisions were required within 4 s, and then the task progressed to the next trial. Participants made their responses using a button box, pressing with their right index finger to indicate whether the target image was on the left or right side of the screen. Participants were encouraged to respond as quickly and accurately as possible. After the decision was made, participants carried out the arrow task again (for 6.5 s minus their response time) followed by a red fixation cross (1 s) indicating the end of the trial. The arrow task served as a non-memory baseline and was included to increase separation of the BOLD signal between trials.

During the fMRI scan, there were four runs of the spatial context task and four runs of the semantic task. Each run contained 36 trials and lasted approximately 6 min. All 72 objects were presented as stimuli across blocks 1 and 2, and across blocks 3 and 4. Each run included 18 decision and 18 no-decision trials. The decision trials in each run were further subdivided into nine SCB and nine MCB decision trials. The decision trials in run 1 and run 2 became the no-decision trials in runs 3 and 4, and vice versa. Spatial context runs preceded semantic runs. The order of trials within each run was counterbalanced between participants. Prior to scanning, participants were given formal instructions for the tasks and shown how to use the response box.

## Study 1: Task-based fMRI

### MRI data acquisition

Whole-brain structural and fMRI data were acquired using a 3T Siemens MRI scanner utilising a 64-channel head coil, tuned to 123 MHz at York Neuroimaging Centre, University of York. A Localiser scan and eight whole-brain functional runs (four of the semantic task, four of the spatial context task) were acquired using a multi-band multi-echo (MBME) EPI sequence, each approximately 6 min long (repetition time [TR] = 1.5 s; echo time [TEs] = 12, 24.83, 37.66 ms; 48 interleaved slices per volume with slice thickness of 3 mm [no slice gap]; FoV = 24 cm [resolution matrix = 3 × 3 × 3; 80×80]; 75° flip angle; 705 volumes per run [235 TRs with each TR collecting 3 volumes]; 7/8 partial Fourier encoding and GRAPPA [acceleration factor = 3, 36 ref lines; multi-band acceleration factor = 2]). Structural T1-weighted images were acquired using an MPRAGE sequence (TR = 2.3 s, TE = 2.26 s; voxel size = 1 × 1 × 1 isotropic; matrix size = 256 × 256, 176 slices; flip angle = 8°; FoV = 256 mm; ascending slice acquisition ordering).

### Multi-echo data pre-processing

This study used an MBME scanning sequence to optimise signal from MT regions (e.g. ATL, MTL) while also maintaining optimal signal across the whole brain (**Halai et al., 2014**). We used TE Dependent ANAlysis (TEDANA, version 0.0.10, https://zenodo.org/records/5461803, https://tedana.readthedocs.io/) to combine the images (**Kundu et al., 2013**; **Posse et al., 1999**). Before images were combined, some pre-processing was performed. FSL_anat (https://fsl.fmrib.ox.ac.uk/fsl/fslwiki/fsl_anat) was used to process the anatomical images, including re-orientation to standard (Montreal Neurological Institute [MNI]) space (fslreorient2std), automatic cropping (robustfov), bias-field correction (RF/B1 – inhomogeneity-correction, using FAST), linear and nonlinear registration to standard-space (using FLIRT and FNIRT), brain extraction (using FNIRT, BET), tissue-type and subcortical structure segmentation (using FAST). The multi-echo data were pre-processed using AFNI (https://afni.nimh.nih.gov/), including de-spiking (3dDespike), slice timing correction (3dTshift; heptic interpolation), and motion correction of all echoes aligned to the first echo (with a cubic interpolation; 3dvolreg was applied to the first echo to realign all images to the first volume; these transformation parameters were then applied to echoes 2 and 3). The pre-processing script is available at OSF (https://osf.io/sh79m/).

## Task-based fMRI data analysis

Further pre-processing of the functional and structural data was carried out using FSL version 6.0 (*Jenkinson et al., 2002*; *Smith et al., 2004*; *Woolrich et al., 2009*). Functional data were pre-processed using FSL's FMRI Expert Analysis Tool (FEAT). The TEDANA outputs (denoised optimally combined time series) registered to the participants' native space were submitted to FSL's FEAT. The first volume of each functional scan was deleted to negate T1 saturation effects. Pre-processing included high-pass temporal filtering (Gaussian-weighted least-squares straight line fitting, with sigma = 50 s), linear co-registration to the corresponding T1-weighted image followed by linear co-registration to MNI152 2 mm standard space (*Jenkinson and Smith, 2001*), which was then further refined using FSL's FNIRT nonlinear registration (*Andersson et al., 2007a*; *Andersson et al., 2007b*) with 10 mm warp resolution, spatial smoothing using a Gaussian kernel with full-width-half-maximum (FWHM) of 5 mm, and grand-mean intensity normalisation of the entire 4D dataset by a single multiplicative factor.

## Task GLM

Second and group-level analyses were also conducted using FSL's FEAT version 6. Pre-processed time series data were modelled using a general linear model (GLM) in FSL, using FILM correcting for local autocorrelation (*Woolrich et al., 2001*). We used an event-related design. Our aim was twofold: (1) to characterise differential activation between the semantic and spatial context tasks at each phase of the trials, and (2) to document any potential differences of activation in response to MCB and SCB trials in probe and decision phases, in each task. To this end, the following eight EVs were entered into a GLM, convolved with a double-gamma haemodynamic response function: the probe, dots and decision phases (only correct responses) were modelled for both MCB and SCB trials (3×2 EVs). Correct decisions made during the arrow task were modelled in a separate EV to use as an explicit baseline. Incorrect and omitted responses in the decision phase, as well as errors made during the arrow task, were combined into a regressor of no interest. The fixation crosses between trials were not explicitly modelled. Probe and dots phases were modelled as fixed-duration epochs, while semantic, spatial context and arrow decisions were modelled using a variable epoch approach, based on each participant's reaction time on that trial. At the first level, the semantic and spatial context tasks were entered into separate models for each run performed by all participants. We then combined all valid runs for each participant into a participant-level analysis, again separately for each task, at the second level (see 'Data exclusions' below for details).

At the group level, we performed two separate univariate analyses. First, we compared activation for the two tasks, contrasting semantic and spatial context models. Inputs for this analysis were lower-level contrasts of the probe phase of each task against the implicit baseline, and the decision phase of each task contrasted against the explicit baseline of arrow decisions. In our second analysis, we used the same lower-level contrasts but examined the semantic and spatial context tasks separately, examining within-task differences between MCB and SCB trials in the probe and decision phases. This also allowed us to explore interactions between MCB/SCB trials and task. We did not include any motion parameters in the model as the data submitted to these first level analyses had already been denoised as part of the TEDANA pipeline (*Kundu et al., 2012*). At the group level, analyses were carried out using FMRIB's local analysis of mixed effects (FLAME1) stage 1 with automatic outlier detection (*Beckmann et al., 2003*; *Woolrich, 2008*; *Woolrich et al., 2004*), using a (corrected) cluster significance threshold of p = 0.05, with a z-statistic threshold of 2.6 (*Eklund et al., 2016*) to define contiguous clusters.

## Data exclusions

We excluded three participants: one due to excessive motion (mean framewise displacement >0.3 mm) in more than 50% of functional runs, another due to misunderstanding the task (0% accuracy in MCB decisions in three out of four runs of the semantic task), and one due to low SCB accuracy in three out of four runs of the spatial context task, with less than 50% of usable data. We also excluded any individual runs where the decision accuracy was equal or below chance level (50%) in the SCB condition. [This threshold was not applied to the MCB condition, which was expected to elicit interference between semantic and spatial context information.] This led to the removal of four runs across three participants in the semantic task, and twelve runs across eight participants in the spatial context task.

Two runs were removed due to data loss (a corrupted EV file and data transfer failure from the MRI scanner). 92.5% of the runs acquired were included in the analysis.

## Study 1: Psychophysiological interaction analysis

In order to test for distinct semantic and spatial memory pathways that connect visual regions to distinct subnetworks of the DMN, we conducted a PPI supplementary analysis. In short, we created semantic and spatial context seeds from the visual regions activated to object and scene probes, and examined their connectivity to the DMN regions activated during the decision phase of the tasks, using two separate models (one for each seed) which examined the main effect of the task. We describe the methods in detail in the relevant section of the supplementary materials (Supplementary analysis: effects of task demands on pathway connectivity).

## Study 1: Representational similarity analysis

Since distinct but adjacent regions were associated with semantic and spatial context decisions, we asked what they represented during probe presentation using RSA. We constructed semantic similarity matrices where trials that shared a specific category (e.g. birds) were assigned the strongest value, and spatial context similarity matrices where pairs of trials belonging to the same room were assigned the strongest value. After single-trial estimation using a least square-single (LSS) approach, we carried out second-order RSA using a searchlight approach to compare semantic and spatial context similarity matrices with neural similarity matrices. This allowed us to identify voxels that were sensitive to semantic and spatial relationships between probes in each of our tasks. We also performed cross-task similarity analysis, correlating semantic similarity to the neural similarity matrix from the spatial context task (and vice versa), to identify regions sensitive to semantic and spatial context information across tasks. We report the methods in detail in the section for this analysis in the supplementary materials (Supplementary analysis: multivariate response to SCB versus MCB).

## Study 2. Passive viewing of objects and scenes

We examined passive viewing of objects and scenes in a sample of 52 healthy volunteers, providing independent ROIs for the analyses of Studies 1 and 3.

### Study 2: Participants

Fifty-two participants with normal, or corrected-to-normal, vision gave informed consent. The study was approved by the Research Ethics Committee at York Neuroimaging Centre.

### Study 2: Stimuli

Dynamic stimuli were 3 s movie clips of faces, bodies, scenes, objects, and scrambled objects (see *Figure 7—figure supplement 2*) designed to localise category-selective visual areas (*Pitcher et al., 2011*). Only the scenes and object stimuli were used in the present study. There were 60 movie clips for each category in which distinct exemplars appeared multiple times. Fifteen different locations were used for the scene stimuli which were mostly pastoral scenes shot from a car window while driving slowly through leafy suburbs, along with films flying through canyons or walking through tunnels that were included for variety. Fifteen different moving objects were selected that minimised any suggestion of animacy of the object itself or of a hidden actor pushing the object (these included mobiles, windup toys, toy planes and tractors, balls rolling down sloped inclines). Within each block, stimuli were randomly selected from within the entire set for that stimulus category (faces, bodies, scenes, objects, scrambled objects).

### Study 2: Procedure and data acquisition

Functional data were acquired over six block-design functional runs lasting 234 s each. Each functional run contained three 18 s rest blocks, at the beginning, middle, and end of the run, during which a series of six uniform colour fields were presented for 3 s. Participants were instructed to watch the movies but were not asked to perform any overt task.

Imaging data were acquired using a 3T Siemens Magnetom Prisma MRI scanner (Siemens Healthcare, Erlangen, Germany) at the University of York. Functional images were acquired with a 20-channel

phased array head coil and a gradient-echo EPI sequence (38 interleaved slices, TR=3 s, TE=30 ms, flip angle = 90%; voxel size 3 mm isotropic; matrix size = 128 × 128) providing whole-brain coverage. Slices were aligned with the anterior to posterior commissure line. Structural images were acquired using the same head coil and a high-resolution T1-weighted 3D fast spoilt gradient (SPGR) sequence (176 interleaved slices, TR=7.8 s, TE=3 ms, flip angle = 20°; voxel size 1 mm isotropic; matrix size = 256 × 256).

## Study 2: Imaging analysis

fMRI data were analysed using AFNI (http://afni.nimh.nih.gov/afni). Images were slice-time corrected and realigned to the third volume of the first functional run and to the corresponding anatomical scan. All data were motion-corrected and any TRs in which a participant moved more than 0.3 mm in relation to the previous TR were discarded from further analysis. The volume-registered data were spatially smoothed with a 4 mm FWHM Gaussian kernel. Signal intensity was normalised to the mean signal value within each run and multiplied by 100 so that the data represented percent signal change from the mean signal value before analysis.

Data from all runs were entered into a GLM by convolving the standard haemodynamic response function with the regressors of interest (faces, bodies, scenes, objects, and scrambled objects) for dynamic and static functional runs. Regressors of no interest (e.g. six head movement parameters obtained during volume registration and AFNI's baseline estimates) were also included in the GLM. Data from all 52 participants were entered in a group whole-brain analysis. Group whole-brain contrasts were generated to quantify the neural responses across the experimental conditions. Scene-selective areas were defined using a contrast of dynamic scenes greater than dynamic objects, and object-selective areas were defined using a contrast of dynamic objects greater than scrambled objects, following convention (*Epstein and Kanwisher, 1998*; *Malach et al., 1995*). Activation maps were calculated using a t-statistical threshold of p=0.001 and a cluster correction of 50 contiguous voxels as these thresholds have been successfully used in other studies to characterise activation in the visual perception literature (e.g. *Nikel et al., 2022*; *Zimmermann et al., 2018*). The whole-brain results are presented in *Figure 2—figure supplement 1*.

## Study 3. Analysis of intrinsic functional connectivity using resting-state fMRI

The results of Study 1 suggested separate visual-DMN pathways recruited by semantic and spatial context tasks. To provide converging evidence for this 'dual pathway' architecture, we examined the intrinsic connectivity of sites identified in the univariate and RSA in a separate sample.

## Study 3: Participants

One hundred and ninety-one student volunteers (mean age = 20.1 ± 2.25 years, range 18–31; 123 females) with normal or corrected-to-normal vision and no history of neurological disorders participated in this study. Written informed consent was obtained from all subjects prior to the resting-state scan. The study was approved by the ethics committees of the Department of Psychology and York Neuroimaging Centre, University of York. This data has been used in previous studies to examine the neural basis of memory and mind-wandering, including ROI-based connectivity analysis and cortical thickness investigations (*Wang et al., 2018a*; *Evans et al., 2020*; *Gonzalez Alam et al., 2018*; *Gonzalez Alam et al., 2019*; *Gonzalez Alam et al., 2022*; *Gonzalez Alam et al., 2021*; *Karapanagiotidis et al., 2017*; *Poerio et al., 2017*; *Turnbull et al., 2019*; *Vatansever et al., 2017*; *Wang et al., 2020*).

## Study 3: Pre-processing

Pre-processing and statistical analyses of resting-state data were performed using the CONN functional connectivity toolbox V.20a (http://www.nitrc.org/projects/conn; *Whitfield-Gabrieli and Nieto-Castanon, 2012*) implemented through SPM (version 12.0) and MATLAB (version 19a). For pre-processing, functional volumes were slice-time (bottom-up, interleaved) and motion-corrected, skull-stripped, and co-registered to the high-resolution structural image, spatially normalised to the MNI space using the unified-segmentation algorithm, smoothed with a 6 mm FWHM Gaussian kernel, and

band-pass filtered (0.008–0.09 Hz) to reduce low-frequency drift and noise effects. A pre-processing pipeline of nuisance regression included motion (twelve parameters: the six translation and rotation parameters and their temporal derivatives), scrubbing (outlier volumes were identified through the composite artefact detection algorithm ART in CONN with conservative settings, including scan-by-scan change in global signal z-value threshold = 3; subject motion threshold = 5 mm; differential motion and composite motion exceeding 95% percentile in the normative sample), and CompCor components (the first five) attributable to the signal from white matter and CSF (*Behzadi et al., 2007*), as well as a linear detrending term, eliminating the need for global signal normalisation (*Chai et al., 2012*; *Murphy et al., 2009*).

### Seed selection and analysis

Intrinsic connectivity seeds were binarised masks derived from: (1) significant univariate clusters; and (2) significant effects identified in RSA. For semantic and spatial probe effects, which characterised effects of visual perception, we created ROIs within *Yeo et al., 2011*, visual central and peripheral networks combined. For semantic and spatial context decisions, we identified regions within *Yeo et al., 2011*, combined DMN subnetworks. We also examined the intrinsic connectivity of regions activated by SCB versus MCB probes in Study 1. For representational similarity analyses, all voxels that survived thresholding at p<0.05 in the MCB conditions for the semantic and context task, as well as the cross-task analyses were binarised and used as seeds. We excluded all non-grey matter voxels that fell within these masks.

### Spatial maps and seed-to-ROI analysis

We performed seed-to-voxel analyses convolved with a canonical haemodynamic response function for each of these seeds. At the group level, analyses were carried out using CONN with cluster correction at p<0.05, and a threshold of p-FDR=0.001 (two-tailed) to define contiguous clusters. Seed-to-ROI connectivity was extracted for each participant and seed using REX implemented in CONN (*Whitfield-Gabrieli and Nieto-Castanon, 2012*), with percentage signal change as units. These values were then entered into a series of repeated-measures ANOVAs.

### Cognitive decoding

Connectivity maps were uploaded to Neurovault (*Gorgolewski et al., 2015*; https://neurovault.org/collections/13821/) and decoded using Neurosynth (*Yarkoni et al., 2011*). Neurosynth is an automated meta-analysis tool that uses text-mining approaches to extract terms from neuroimaging articles that typically co-occur with specific peak coordinates of activation. It can be used to generate a set of terms frequently associated with a spatial map. The results of cognitive decoding were rendered as word clouds using in-house scripts implemented in R. We excluded terms referring to neuroanatomy (e.g. 'inferior' or 'sulcus'), as well as the second occurrence of repeated terms (e.g. 'semantic' and 'semantics'). The size of each word in the word cloud relates to the frequency of that term across studies.

## Structural connectivity analysis

To provide converging evidence for parallel visual-to-DMN pathways, we performed tractography analysis using DTI data from an independent sample derived from the HCP.

### DTI pre-processing

We used data from a subgroup of 164 HCP participants who underwent diffusion-weighted imaging at 3 T (*Uğurbil et al., 2013*; http://www.humanconnectome.org/study/hcp-young-adult/). The imaging parameters were previously described in *Uğurbil et al., 2013*, involved acquiring 111 near-axial slices with an acceleration factor of 32, an isotropic resolution of 1.25 mm$^3$, and coverage of the entire head. The diffusion-weighted images were obtained using 90 uniformly distributed gradients in multiple Q-space shells (*Caruyer et al., 2013*), and this process was repeated three times with different b-values and phase-encoding directions. We used a pre-processed version of this dataset, previously described (*Karolis et al., 2019*; *Thiebaut de Schotten et al., 2020*; *Vu et al., 2015*), that included steps to correct for susceptibility-induced off-resonance field, motion, and geometrical distortion.

We used StarTrack software (https://www.mr-startrack.com) to perform whole-brain deterministic tractography in the native DWI space. We applied an algorithm for spherical deconvolutions (damped Richardson-Lucy), with a fixed fibre response corresponding to a shape factor of $\alpha=1.5 \times 10^{-3}$ mm$^2$·s$^{-1}$ and a geometric damping parameter of 8. We ran 200 algorithm iterations. The absolute threshold was set at three times the spherical fibre orientation distribution (FOD) of a grey matter isotropic voxel, and the relative threshold was set at 8% of the maximum amplitude of the FOD (*de Schotten et al., 2011*). To perform the whole-brain streamline tractography, we used a modified Euler algorithm (*Dell'Acqua et al., 2013*) with an angle threshold of 45°, a step size of 0.625 mm, and a minimum streamline length of 15 mm.

To standardise the structural connectome data, we followed these steps: first, we converted the whole-brain streamline tractography into streamline density volumes, with the intensity corresponding to the number of streamlines crossing each voxel. Second, we generated a study-specific template of streamline density volumes using the Greedy symmetric diffeomorphic normalisation pipeline provided by ANTs. This average template was created for all subjects. Third, we co-registered the template with a standard 1 mm MNI152 template using the FLIRT tool in FSL to produce a streamline density template in the MNI152 space. Finally, we registered individual streamline density volumes to the template and applied the same transformation to the individual whole-brain streamline tractography using ANTs GreedySyn and the Trackmath tool in the Tract Querier software package (*Wassermann et al., 2016*). This produced whole-brain streamline tractography in the standard MNI152 space.

## Tract extractions and ROI analysis

Our starting point for extracting semantic and spatial context pathway tracts was each participant's whole-brain streamline tractography in MNI (1 mm) space. We used the same univariate regions described in Section 2.3.3 as seeds (i.e. the seeds in the intrinsic connectivity analysis): these consisted of regions that were activated during the probe phase of each task, masked by Yeo's visual networks, and regions that were activated during the decision phase of each task, masked by Yeo's DMN. For each of our seeds, we used Trackvis (*Wang and Benner, 2007*) to extract all streamlines emerging from these regions as a volume, yielding one streamline group per seed per participant. Then, for each probe visual seed, we calculated what percentage of streamlines touched one decision DMN ROI or the other (activated by semantic and spatial context decisions; percentages adding to 100%); likewise, for each decision DMN seed, we calculated what percentage of streamlines touched either visual probe ROI (activated by object and scene probes; again adding to 100%). This allowed us to examine if the object probe regions were more connected to the semantic decision DMN regions, and if the scene probe regions were more connected to the spatial context DMN regions, in line with dual pathways.

## Situating the pathways in whole-brain gradients

We examined the position of the semantic and context pathways in a functional connectivity space defined by the first two dimensions of whole-brain intrinsic connectivity patterns, frequently referred to as 'gradients'. The first dimension of this space relates to the distinction between the connectivity patterns of unimodal and heteromodal cortical regions, while the second dimension captures the separation of visual and auditory/somatomotor regions (*Margulies et al., 2016*). This analysis can reveal whether the semantic pathway shows more of a balance between visual and somatosensory/auditory modalities than the spatial context pathway, in line with view that concepts are heteromodal, abstracted from multiple sensory-motor features (*Ralph et al., 2017*). The analysis can also show whether the spatial context pathway is anchored in more visual portions of this functional space, in line with this modality's importance for scene processing (*Epstein and Baker, 2019*).

First, we examined the univariate BOLD activation for each participant during the probe and decision phases of each task. The decision phase was contrasted with the arrow task baseline to control for low-level motor responses. Next, we identified the MNI voxel location of the peak response for each participant: for activation during the decision phase, this was done within a mask of the DMN from the *Yeo et al., 2011*, 7-network parcellation, while for probe responses, we performed this analysis within the visual network of the same parcellation. We then fitted a sphere with a 5 mm radius around this peak and used it as a ROI to extract the mean value in *Margulies et al., 2016* maps for the two dimensions or gradients described above. The results were entered into a repeated-measures 2×2

ANOVA with task and gradient as factors to establish whether the semantic and spatial context pathways differed in their location in this functional space.

### ROI-based ANOVAs

ROI-based analyses of activation and intrinsic connectivity in Studies 1 and 3 were performed using FSL's 'Featquery' tool for Study 1 and REX for Study 3, which we used to extract the percentage signal change within unweighted, binarised masks. The ANOVAs were carried out using IBM SPSS Statistics version 27. The results of post hoc tests to interpret significant interactions were corrected for multiple comparisons using the Holm-Bonferroni method (*Aickin and Gensler, 1996*). All the p values reported in the Results section are Holm-Bonferroni adjusted p values.

### Visualisations of neural results

Brain maps were produced in BrainNet (*Xia et al., 2013*) using the extremum voxel algorithm, with the exception of slices depicted in *Figures 2–4*, which were produced in FSL Eyes (*Figures 2 and 3*) and MRIcroGL (*Figure 4*). Maps are provided in the following Neurovault collection: https://neurovault.org/collections/13821/.

## Acknowledgements

This project was funded by the European Research Council (ERC) under the European Union's Horizon 2020 research and innovation programme (Project ID: 771863 – FLEXSEM to EJ; Project ID: 818521 – DISCONNECTOME to MTS; Project ID: 866533 to DSM). Additionally, this work was conducted in the framework of the University of Bordeaux's IHU 'Precision & Global Vascular Brain Health Institute – VBHI', IdEx 'Investments for the Future' program RRI 'IMPACT', which received financial support from the France 2030 program.

## Additional information

### Funding

| Funder | Grant reference number | Author |
|---|---|---|
| European Research Council | Project ID: 771863 - FLEXSEM | Elizabeth Jefferies |
| European Research Council | Project ID: 818521 - DISCONNECTOME | Michel Thiebaut de Schotten |
| European Research Council | Project ID: 866533 | Daniel S Margulies |

The funders had no role in study design, data collection and interpretation, or the decision to submit the work for publication.

### Author contributions

Tirso RJ Gonzalez Alam, Conceptualization, Resources, Data curation, Software, Formal analysis, Investigation, Visualization, Methodology, Writing – original draft, Project administration, Writing – review and editing; Katya Krieger-Redwood, Data curation, Formal analysis, Methodology; Dominika Varga, Conceptualization, Resources, Data curation, Formal analysis, Investigation, Methodology, Writing – original draft; Zhiyao Gao, Conceptualization, Resources, Data curation, Formal analysis, Investigation, Methodology, Writing – original draft, Writing – review and editing; Aidan J Horner, Tom Hartley, Conceptualization, Writing – review and editing; Michel Thiebaut de Schotten, Resources, Formal analysis, Investigation, Methodology, Writing – original draft, Writing – review and editing; Magdalena Sliwinska, David Pitcher, Resources, Formal analysis, Investigation, Writing – review and editing; Daniel S Margulies, Conceptualization, Resources, Writing – review and editing; Jonathan Smallwood, Conceptualization, Resources, Supervision, Writing – review and editing; Elizabeth Jefferies, Conceptualization, Resources, Formal analysis, Supervision, Funding acquisition, Investigation, Visualization, Methodology, Writing – original draft, Project administration, Writing – review and editing

## Author ORCIDs
Tirso RJ Gonzalez Alam (ID) https://orcid.org/0000-0003-4510-2441
Zhiyao Gao (ID) https://orcid.org/0000-0002-8909-8096
Aidan J Horner (ID) https://orcid.org/0000-0003-0882-9756
Michel Thiebaut de Schotten (ID) https://orcid.org/0000-0002-0329-1814

## Ethics

Human subjects: Ethical approval was obtained from the Research Ethics Committees of the Department of Psychology and York Neuroimaging Centre, University of York (P1391). Written informed consent was obtained from all subjects prior to testing.

Reviewer #2 (Public review): https://doi.org/10.7554/eLife.94902.3.sa1
Author response https://doi.org/10.7554/eLife.94902.3.sa2

---

# Additional files

## Supplementary files
MDAR checklist

Supplementary file 1. Cluster Information for Neuroimaging Results from Study 1.

Supplementary file 2. Paired t-tests contrasting spatial similarity of participant-level activation with group-level context and semantic pathways and non-pathways.

## Data availability

The scripts used in the presentation of the task, the analysis of the neuroimaging data and the visualisation of the results reported here can be consulted in the OSF collection associated with this paper (https://osf.io/sh79m/). We do not have sufficient consent for the public release of individual-level data; researchers wanting access to these data should contact the Research Ethics Committee of the York Neuroimaging Centre (rec-submission@ynic.york.ac.uk). Data will be released when this is possible under the terms of the UK and EU General Data Protection Regulations. Group-level brain maps used to produce the figures are available from the following Neurovault collection: https://neurovault.org/collections/13821/.

The following dataset was generated:

| Author(s) | Year | Dataset title | Dataset URL | Database and Identifier |
|---|---|---|---|---|
| Gonzalez-Alam TR, Jefferies B | 2023 | Visual to default network pathways: A double dissociation between semantic and spatial cognition | https://osf.io/sh79m/ | Open Science Framework, sh79m |

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

# Appendix 1

## Supporting materials

### Supplementary analysis: resting-state maps with stricter thresholding

Since the pattern of results obtained from resting-state analysis can be sensitive to threshold decisions, we reproduced the group-level maps depicted in *Figures 2E and 3E* using a stricter threshold. We originally thresholded these results using the default of the CONN software package (cluster-forming threshold of p=0.05, equivalent to T=1.65). For increased rigour, we reproduced the thresholded maps from *Figures 2E and 3E* further increasing the threshold from p=0.05, equivalent to T=1.65, to p=0.001, equivalent to T=3.1. The resulting maps were very similar, showing minimal change, with a spatial correlation of r>0.99 between the strict and lax threshold versions of the maps for both the probe and decision seeds. These maps can be downloaded from the OSF collection associated with this project.

### Supplementary analysis: eroded masks replication analysis

The proximity of visual peripheral and DMN-C network borders is a property of the organisation of these networks (*Silson et al., 2019*; *Steel et al., 2021*). However, this could give rise to the potential for spatial mixing of the resting-state signal during intrinsic connectivity analysis due to smoothing, falsely inflating the strength of connectivity, since our visual and DMN masks for the spatial context task showed spatial adjacency. To address this concern, we re-analysed the resting-state data presented in *Figure 4b and c* (connectivity from DMN decision regions to visual probe regions and vice versa) by eroding the visual probe and DMN decision ROIs for the spatial context task using fslmaths. We eroded the masks until the smallest gap between them exceeded the size of our 6 mm FWHM smoothing kernel, which eliminates the potential for spatial mixing of signals due to ROI adjacency. The eroded ROIs can be consulted in the OSF collection associated with this project. We repeated the ANOVAs associated with these analyses. The results, presented in *Figure 4—figure supplement 1* below, confirmed the pattern of findings reported in the main analysis. We did not erode the respective ROIs for the semantic task, given that adjacency is not an issue for the ROIs derived from that task.

The visual-to-DMN ANOVA showed main effects of seed (F(1,190)=22.82, p<0.001), ROI (F(1,190)=9.48, p=0.002) and a seed by ROI interaction (F(1,190)=67.02, p<0.001). Post hoc contrasts confirmed there was stronger connectivity between object probe regions and semantic versus spatial context decision regions (t(190)=3.38, p<0.001), and between scene probe regions and spatial context versus semantic decision regions (t(190)=–7.66, p<0.001).

The DMN-to-visual ANOVA confirmed this pattern: again, there was a main effect of ROI (F(1,190)=4.3, p=0.039) and a seed by ROI interaction (F(1,190)=57.59, p<0.001), with post hoc contrasts confirming stronger intrinsic connectivity between DMN regions implicated in sematic decisions and object probe regions (t(190)=5.06, p<0.001), and between DMN regions engaged by spatial context decisions and scene probe regions (t(190)=3.25, p=0.001).

### Supplementary analysis: replicating resting-state connectivity pathways with task-independent ROIs

We performed a supplementary analysis using task-independent ROIs to confirm that the intrinsic connectivity-based pathways could be identified even when the seeds and ROIs were not derived from the same task. In this analysis, we used the same seeds as the main analysis (*Figure 4a*; the conjunction of semantic and context decision activation within DMN and object and scene probes within visual networks), while the ROIs were either visual localiser masks for objects and scenes, or DMN subsystems from the *Yeo et al., 2011*, parcellation. The results can be consulted in *Figure 4—figure supplement 2*. The DMN-to-visual ANOVA revealed main effects of DMN seed (F(1,190)=137.72, p<0.001), visual ROI based on the localisers from Study 2 (F(1,190)=23.87, p<0.001) and their interaction (F(1,190)=20.92, p<0.001). Post hoc tests revealed that DMN regions associated with spatial context decisions showed stronger connectivity to both visual regions associated with viewing scenes, and visual regions associated with viewing objects, relative to DMN regions associated with semantic decisions (scenes: t(190)=11.57, p<0.001; objects: t(190)=7.35, p<0.001). The significant interaction term reveals that this difference between context and semantic seeds was more pronounced for the scene than the object localiser regions, confirming a greater importance of the context pathway for making decisions based on visual scene information.

The visual-to-DMN ANOVA revealed main effects of visual seed (F(1,190)=33.98, p<0.001), DMN ROI (F(1,190)=65.46, p<0.001) and their interaction (F(2,380)=119.14, p<0.001). Post hoc tests revealed that visual regions associated with viewing object probes showed stronger connectivity to FT-DMN regions relative to regions associated with viewing spatial context probes (t(190)=3.22, p=0.002), while regions associated with viewing spatial context probes showed stronger connectivity to core and MT-DMN regions than regions associated with viewing object probes (core DMN: t(190)=4.41, p<0.001; MT-DMN: t(190)=11.7, p<0.001).

## Supplementary analysis: replicating pathways' structural connectivity from the DMN end

To confirm dissociable pathways based on structural connectivity can be identified not only when examining visual-to-DMN regions, but also the reverse, we performed a supplementary analysis seeding the DMN, and examined the strength of connections to visual ROIs (defined using activation to object and scene probes in Study 1 masked by visual networks). The results can be consulted on *Figure 4—figure supplement 3*. The DMN-to-visual ANOVA showed a main effect of visual ROI (F(1,163)=506.5, p<0.001) and a seed by ROI interaction (F(1,163)=215.27, p<0.001). Post hoc tests revealed that both DMN seeds, associated with semantic and spatial context decisions, showed stronger connectivity to visual regions responding more to scenes than to objects. However, this connectivity difference was greater for DMN regions activated by spatial context than semantic decisions (semantic DMN: t(163)=3.92, p<0.001; spatial context DMN: t(163)=2381.12, p<0.001).

## Supplementary analysis: effects of task demands on pathway connectivity
### Methods

In order to test for distinct semantic and spatial memory pathways that connect visual regions to distinct subnetworks of the DMN, we conducted a PPI analysis. Semantic and spatial context visual seeds were created from the univariate activation to object and scene probes in the semantic and spatial tasks respectively, masked by *Yeo et al., 2011*, 7-network parcellation visual network. The time series of these seeds were then extracted after pre-processing. We then ran two separate models (one for each seed), which examined the main effect of the experimental condition (i.e. *SCB trials of the semantic task, MCB trials of the semantic task, SCB trials of the spatial context task, and MCB trials of the spatial context task*). These models included all eight regressors from the basic task model of Study 1 described in Section: Task GLM, a PPI term for each of the seven conditions and phases of the task (SCB/MCB trials of the probe, dots and decision phases, and the arrow task), as well as the time series of the visual probe seeds, using the generalised psychophysiological interaction approach (*McLaren et al., 2012*). The regressors were not orthogonalised. All runs of each task were combined using fixed-effects analyses for each participant, which allowed us to extract the connectivity parameters for each experimental condition for each participant in each seed model.

### Results and discussion

We examined how connectivity within the pathways changes depending on task demands in a PPI analysis. We took the visual regions showing differential activation to object and scene probes as seeds (shown in *Figures 2e and 4a*), while the ROIs were regions sensitive to semantic and spatial context decisions within the DMN (shown in *Figures 3e and 4a*). We anticipated that the scene probe regions would increase their connectivity to spatial context decision regions during the decision phase of the spatial context task, whilst the object probe regions would increase their connectivity to semantic decision regions during the decision phase of the semantic task. A repeated-measures ANOVA including task (semantic/spatial context), seed (object/scene probe), ROI (semantic/spatial context decision), and condition (SCB/MCB) as factors revealed two-way interactions for task by seed (F(1,26)=10.85, p=0.003) and seed by ROI (F(1,26)=8.57, p=0.007), as well as a three-way interaction for task by seed by ROI (F(1,26)=5.2, p=0.031). Since we found no effect of condition, we averaged across this factor for the following analyses. The results are shown in *Figure 4—figure supplement 4*. To understand the three-way interaction, separate two-way ANOVAs using seed (object/scene probe regions) and ROI (semantic/spatial context decision regions) as factors were computed for the spatial context and semantic tasks. The semantic task showed a main effect of seed (F(1,26)=6.97, p=0.014), but no effect of ROI or interaction: the object seed was more connected to both semantic and spatial context DMN decision regions during the semantic task. The spatial context task showed a main effect of seed (F(1,26)=5.89, p=0.022), and a seed by ROI

interaction (F(1,26)=10.25, p=0.004). Post hoc t-tests showed that the scene probe regions were more connected to spatial context decision regions during the spatial context task than object probe regions (t(26)=3.52, p=0.002). In contrast, there was no difference in connectivity between these two seeds and the semantic decision regions.

In sum, the PPI models characterised how inputs to the pathways are flexibly configured to suit our current goals. The visual ends of the pathways showed opposing patterns of connectivity to spatial context DMN regions depending on the task.

## Supplementary analysis: individual location of pathways in whole-brain gradients

Our analysis of the location of the pathways in whole-brain gradient connectivity space showed that peak responses during semantic decisions occurred in more abstract, less visual regions of the DMN relative to spatial context decisions. However, the scatterplots in the top panel of *Figure 5* do not allow to distinguish whether these effects took place at the individual level, since the data points are not linked across tasks. In light of this, we plotted the same data comparing the gradient values for the peak responses in each of our tasks at the participant level. The peaks for each participant across tasks are linked with a line. Cases where the pattern was reversed are highlighted with dashed lines (7/27 participants in each gradient, see *Figure 5—figure supplement 1*). This analysis showed that in the majority of cases, at the individual level, the pattern of group-level results held.

## Supplementary analysis: multivariate response to SCB versus MCB

### Methods

Since distinct but adjacent regions were associated with semantic and spatial context decisions, we asked what they represented during probe presentation in MCB and SCB trials using RSA. Probes were presented across decision and no-decision trials, allowing a large number of probe responses to be included in the analysis. We examined the voxels that responded to contrasts between semantic and spatial context decisions (including all significant, suprathreshold voxels at the group level within a single ROI). We constructed semantic similarity matrices for each participant using all pairs of trials from the semantic task, encoding category similarity on a scale of 0–2. Pairs of trials that shared a specific category (e.g. birds) were assigned the strongest value (2), while those that shared only their superordinate category (animals versus man-made objects) were assigned the middle value (1); pairs of trials from different superordinate categories were assigned the weakest value (0). We also constructed spatial context similarity matrices for each participant, encoding the relationships between rooms and buildings on a scale of 0–2. Pairs of trials belonging to the same room were assigned the strongest value (2), while those belonging to different rooms of the same building were assigned a middle value (1); trials that belonged to different buildings were assigned the lowest value (0).

### Single-trial estimation

GLMs were performed separately to estimate the activation pattern for each of 144 trials during the probe phase in the two tasks. An LSS approach was used, in which the trial of interest was modelled as one regressor, with all other trials modelled as separate regressors (*Mumford et al., 2012*). These models included eight regressors: (1) the probe phase of interest (SCB or MCB); (2 and 3) all other probe phases (SCB and MCB); (4) dots SCB; (5) dots MCB; (6) decision SCB; (7) decision MCB; (8) arrow trials. Since the analysis focussed on probe presentation rather than decisions, incorrect trials were not excluded. Each event was modelled at the time of stimulus onset and convolved with a canonical haemodynamic response function (double gamma), whereas the fixations were treated as an implicit baseline. Pre-whitening was applied. The same pre-processing procedure as in the univariate analysis was used except that no spatial smoothing was applied. This voxel-wise GLM was used to compute the activation associated with each trial, using the t-map for RSA to increase reliability by normalising for noise (*Walther et al., 2016*).

### Second-order representational similarity analysis

A searchlight approach compared semantic and spatial context similarity matrices with neural similarity matrices. Neural pattern similarity was estimated for cubic ROIs within t-maps for each trial, containing 125 voxels surrounding a central voxel (*Fairhall and Caramazza, 2013*; *Gao et al., 2022*; *Leshinskaya et al., 2017*; *Malone et al., 2016*; *Stolier and Freeman, 2016*; *Viganò and*

*Piazza, 2020*; *Wang et al., 2018b*). In each of these cubes, we derived a neural similarity matrix from Pearson correlations of pairs of trials. We excluded any pairs presented in the same run to avoid any autocorrelation. Spearman's rank correlation was used to measure the alignment between task and brain-based models during the probe phase. The resulting coefficients were Fisher's z-transformed and then entered into a group-level analysis carried out using FSL's Randomise (*Anderson and Robinson, 2001*; *Winkler et al., 2014*) (5000 permutations with threshold-free cluster enhancement), thresholding the results at p<0.05.

We also performed cross-task similarity analysis, correlating semantic similarity to the neural similarity matrix from the spatial context task (and vice versa). If participants use semantic information learned during training to guide spatial context decisions, or spatial context information from training to facilitate semantic decisions, we might be able to identify regions sensitive to semantic and spatial context information across tasks. This should only be the case in SCB and not MCB trials.

## Results and discussion

The univariate analysis in the main text shows that when there is no alignment between spatial context and semantic information (in MCB trials), the heteromodal areas that are activated by the task show higher pathway-specific connectivity. In contrast, when information integration across space and meaning is facilitated by the structure of the task, spatial context trials show more activation in regions with lower connectivity to the spatial context pathway, but higher connectivity to the semantic pathway. In this way, right angular gyrus was found to have a potential role in integrating the visual-to-DMN pathways.

A follow-up analysis used a multivariate approach to establish how neural patterns related to the task reflected information integration. We performed RSA using a searchlight approach within a mask that combined semantic and spatial context task decision maps, using data acquired during the probe phase (since there were more probe than decision time-points). This method allowed us to select regions sensitive to semantic and spatial context information, while ensuring that the search space was not derived from the same data used for the RSA. First, we asked if we could detect regions sensitive to category during the semantic task and sensitive to location during the spatial context task, in the MCB trials. There were regions that represented semantic and spatial context similarity in bilateral and left LOC respectively (*Figure 6—figure supplement 1a*). Next, we performed a cross-task RSA in the SCB trials to identify areas that represented information relevant to one task in the other (e.g. areas that represented semantic information during the spatial context task and vice versa). The results of this analysis revealed right LOC regions that captured spatial context information during the semantic task (*Figure 6—figure supplement 1b*). No medial regions were found in these analyses.

Finally, we investigated the intrinsic connectivity of these multivariate clusters to the semantic and spatial context pathways (*Figure 4d*, *Figure 6—figure supplement 1c*), to establish whether cross-task RSA regions thought to support integration have an intermediate pattern of connectivity to both pathways. We used the MCB semantic and spatial context RSA clusters and the cross-task RSA result from *Figure 6—figure supplement 1a–c* as seeds in a seed-to-ROI analysis of intrinsic connectivity using independent data from Study 3. We performed a 2 two-way repeated-measures ANOVA, using a 3×2 design, entering seed (semantic, spatial context and cross-similarity RSA results), and pathway (ROIs in *Figure 6—figure supplement 1c*) as factors. The results can be seen in *Figure 6—figure supplement 1d*. There were main effects of seed (F(1.54,293.32)=194.24, p<0.001), ROI (F(1,190)=290.07, p<0.001) and their interaction (F(1.58,300.36)=123.36, p<0.001). Post hoc comparisons confirmed that the spatial context pathway was equally connected to the spatial context RSA cluster and to the cross-task RSA cluster (spatial context > cross-task: t(190)=–0.155, p>0.05), with both of these clusters being significantly more connected than the semantic RSA cluster (spatial context > semantic: t(190)=3.34, p=0.002; cross-task > semantic: t(190)=4.41, p<0.001). The semantic pathway was most connected to the semantic RSA cluster, less connected to the cross-task RSA cluster, and least connected to the spatial context RSA cluster (semantic > cross-task: t(190)=9.2, p<0.001; cross-task > spatial context: t(190)=16.31, p<0.001). In this way, the cross-task representation of spatial context information in visual regions during the semantic task showed an intermediate pattern of connectivity (particularly to the semantic pathway).

These cross-talk regions, in right LOC, have been implicated in the integration of objects with their spatial location, allowing object coherence in space in the face of saccadic movements that

occur in natural vision while navigating environments (*McKyton and Zohary, 2007*). Another fMRI study investigating the structure of identity-related and location-related representations in visual regions found an interaction effect of these aspects of knowledge for objects positioned in expected spatial locations, in a similar fashion to our study (*Gronau et al., 2008*).

For completion, we present the results of RSA of SCB trials in *Figure 6—figure supplement 2* below.

## Study 1. Training and test materials

In the training session, participants navigated virtual environments populated with objects, with the aim of learning these objects' location in the environments. During this training, their memory was tested in a two-alternative forced-choice test and a matching task. *Figure 7—figure supplement 1* shows some example screenshots from videos used by participants to learn the environments, and the tests used during the training session to probe participants' memory.

## Study 2: Materials and results

*Figure 7—figure supplement 2* shows example stimuli used in Study 2. Our localisers focussed on the scenes and objects conditions.

As shown in the left panel of *Figure 2—figure supplement 1*, the objects over scrambled objects contrast revealed activation in areas associated with the processing of objects and grasping, such as bilateral LOC, fusiform, parietal and precuneal cortex. The scenes over objects contrast showed activation in medial occipital and retrosplenial cortex, associated with scene processing. Since we were interested in using these activation maps as masks to determine relevant regions of the visual end of our pathways, we masked these effect maps by large-scale subnetworks implicated in vision from an influential parcellation (*Yeo et al., 2011*; visual central and visual peripheral networks combined). The resulting masks showed some voxels in common, and since the aim of this analysis was to identify areas that preferentially respond to objects or scenes, we excluded these voxels in a final step (i.e. we removed all voxels from the objects mask that were also part of the scenes mask, and vice versa). These results can be consulted on the right panel of *Figure 2—figure supplement 1*.

