## [Editor Report · eLife assessment]

This **useful** experiment seeks to better understand how memory interacts with incoming visual information to effectively guide human behavior. Using several methods, the authors identify two distinct pathways relating visual processing to the default mode network: one that emphasizes semantic cognition, and the other, spatial cognition. The evidence presented is **solid** and will be of interest to cognitive and systems neuroscientists.

---

## [Referee Report · Reviewer #2 (Public review)]

Summary:

In this manuscript, Gonzalez Alam et al. sought to understand how memory interacts with incoming visual information to effectively guide human behavior by using a task that combines spatial contexts (houses) with objects of one or more other semantic categories. Three additional datasets (all from separate participants) were also employed: one that functionally localized regions of interest (ROIs) based on subtractions of different visually presented category types (in this case, scenes, objects, and scrambled objects); another consisting of resting-state functional connectivity scans, and a section of the Human Connectome Project that employed DTI data for structural connectivity analysis. Across multiple analyses, the authors identify dissociations between regions preferentially activated during scene or other object judgments, between the functional connectivity of regions demonstrating such preferences, and in the anatomical connectivity of these same regions. The authors conclude that the processing streams that take in visual information and support semantic or spatial processing are largely parallel and distinct.

Strengths:

(1) Recent work has reconceptualized the classic default mode network as parallel and interdigitated systems (e.g., Braga & Buckner, 2017; DiNicola et al., 2021). The current manuscript is timely in that it attempts to describe how information is differentially processed by two streams that appear to begin in visual cortex and connect to different default subnetworks. Even at a group level where neuroanatomy is necessarily blurred across individuals, these results provide clear evidence of stimulus-based processing dissociation.

(2) The manuscript analyzes data from multiple independent datasets. It is therefore unlikely that a single experimenter choice in any given analysis would spuriously produce the general convergence of the results reported in this manuscript.

Weaknesses:

(1) The manuscript makes strong distinctions between spatial processing and other forms of semantic processing. However, it is not clear if scenes are uniquely different from other stimulus categories, such as faces or tools. As is noted by the authors in their revised discussion section, the design of the experiment does not allow for a category-level generalization beyond scenes. The dichotomization of semantic and spatial information invoked throughout the manuscript should be read with this limitation in mind.

(2) Although the term "objects" is used by the authors to refer to the stimuli placed in scenes, it is a mixture of other stimulus categories, including various types of animals, tools, and other manmade objects. Different regions along the ventral stream are thought to process these different types of stimuli (e.g., Martin, 2007, Ann Rev Psychol), but as they are not being modeled separately, the responses associated with "object" processing in this manuscript are necessarily blurring across known distinctions in functional neuroanatomy.

---

## [Author Response]

The following is the authors’ response to the original reviews.

**Public Reviews:**

**Reviewer #1 (Public Review):**
In this study, Gonzalez Alam et al. report a series of functional MRI results about the neural processing from the visual cortex to high-order regions in the default-mode network (DMN), compiling evidence from task-based functional MRI, resting-state connectivity, and diffusionweighted imaging. Their participants were first trained to learn the association between objects and rooms/buildings in a virtual reality experiment; after the training was completed, in the task-based MRI experiment, participants viewed the objects from the earlier training session and judged if the objects were in the semantic category (semantic task) or if they were previously shown in the same spatial context (spatial context task). Based on the task data, the authors utilised resting-state data from their previous studies, visual localiser data also from previous studies, as well as structural connectivity data from the Human Connectome Project, to perform various seed-based connectivity analysis. They found that the semantic task causes more activation of various regions involved in object perception while the spatial context task causes more activation in various regions for place perception, respectively. They further showed that those object perception regions are more connected with the frontotemporal subnetwork of the DMN while those place perception regions are more connected with the medial-temporal subnetwork of the DMN. Based on these results, the authors argue that there are two main pathways connecting the visual system to highlevel regions in the DMN, one linking object perception regions (e.g., LOC) leading to semantic regions (e.g., IFG, pMTG), the other linking place perception regions (e.g., parahippocampal gyri) to the entorhinal cortex and hippocampus.Below I provide my takes on (1) the significance of the findings and the strength of evidence, (2) my guidance for readers regarding how to interpret the data, as well as several caveats that apply to their results, and finally (3) my suggestions for the authors.(1) Significance of the results and strength of the evidenceI would like to praise the authors for, first of all, trying to associate visual processing with high-order regions in the DMN. While many vision scientists focus specifically on the macroscale organisation of the visual cortex, relatively few efforts are made to unravel how neural processing in the visual system goes on to engage representations in regions higher up in the hierarchy (a nice precedent study that looks at this issue is by Konkle and Caramazza, 2017). We all know that visual processing goes beyond the visual cortex, potentially further into the DMN, but there's no direct evidence. So, in this regard, the authors made a nice try to look at this issue.

We thank the reviewer for their positive feedback and for their very thoughtful and thorough comments, which have helped us to improve the quality of the paper.

Having said this, the authors' characterisation of the organisation of the visual cortex (object perception/semantics vs. place perception/spatial contexts) does not go beyond what has been known for many decades by vision neuroscience. Specifically, over the past two decades, numerous proposals have been put forward to explain the macroscale organisation of the visual system, particularly the ventrolateral occipitotemporal cortex. A lateral-medial division has been reliably found in numerous studies. For example, some researchers found that the visual cortex is organised along the separation of foveal vision (lateral) vs. peripheral vision (medial), while others found that it is structured according to faces (lateral) vs. places (medial). Such a bipartite division is also found in animate (lateral) vs. inanimate (medial), small objects (lateral) vs. big objects (medial), as well as various cytoarchitectonic and connectomic differences between the medial side and the lateral side of the visual cortex. Some more recent studies even demonstrate a tripartite division (small objects, animals, big objects; see Konkle and Caramazza, 2013). So, in terms of their characterisation of the visual cortex, I think Gonzalez Alam et al. do not add any novel evidence to what the community of neuroscience has already known.

The aim of our study was not to provide novel evidence about visual organisation, but rather to understand how these well-known visual subdivisions are related to functional divisions in memory-related systems, like the DMN. We agree that our study confirms the pattern observed by numerous other studies in visual neuroscience.

However, the authors' effort to link visual processing with various regions of the DMN is certainly novel, and their attempt to gather converging evidence with different methodologies is commendable. The authors are able to show that, in an independent sample of restingstate data, object-related regions are more connected with semantic regions in the DMN while place-related regions are more connected with navigation-related regions in the DMN, respectively. Such patterns reveal a consistent spatial overlap with their Kanwisher-type face/house localiser data and also concur with the HCP white-matter tractography data. Overall, I think the two pathways explanation that the authors seek to argue is backed by converging evidence. The lack of travelling wave type of analysis to show the spatiotemporal dynamics across the cortex from the visual cortex to high-level regions is disappointing though because I was expecting this type of analysis would provide the most convincing evidence of a 'pathway' going from one point to another. Dynamic caudal modelling or Granger causality may also buttress the authors' claim of pathway because many readers, like me, would feel that there is not enough evidence to convincingly prove the existence of a 'pathway'.

By ‘pathway’ we are referring to a pattern of differential connectivity between subregions of visual cortex and subregions of DMN, suggesting there are at least two distinct routes between visual and heteromodal regions. However, these routes don’t have to reflect a continuous sequence of cortical areas that extend from visual cortex to DMN – and given our findings of structural connectivity differences that relate to the functional subdivisions we observe, this is unlikely to be the sole mechanism underpinning our findings. We have now clarified this in the discussion section of the manuscript. We agree it would be interesting to characterise the spatiotemporal dynamics of neural propagation along our pathways, and we have incorporated this proposal into the future directions section.

“One important caveat is that we have not investigated the spatiotemporal dynamics of neural propagation along the pathways we identified between visual cortex and DMN. The dissociations we found in task responses, intrinsic functional connectivity and white matter connections all support the view that there are at least two distinct routes between visual and heteromodal DMN regions, yet this does not necessarily imply that there is a continuous sequence of cortical areas that extend from visual cortex to DMN – and given our findings of structural connectivity differences that relate to the functional subdivisions we observe, this is unlikely to be the sole mechanism underpinning our findings. It would be interesting in future work to characterise the spatiotemporal dynamics of neural propagation along visualDMN pathways using methods optimised for studying the dynamics of information transmission, like Granger causality or travelling wave analysis.”

We have also edited the wording of sentences in the introduction and discussion that we thought might imply directionality or transmission of information along these pathways, or to clarify the nature of the pathways (please see a couple of examples below):

In the Introduction:

“We identified dissociable pathways of connectivity between from different parts of visual cortex to and DMN subsystems “

In the Discussion:

“…pathways from visual cortex to DMN -> …pathways between visual cortex and DMN“.

(2) Guidance to the readers about interpretation of the dataThe organisation of the visual cortex and the organisation of the DMN historically have been studied in parallel with little crosstalk between different communities of researchers. Thus, the work by Gonzalez Alam et al. has made a nice attempt to look at how visual processing goes beyond the realm of the visual cortex and continues into different subregions of the DMN.

While the authors of this study have utilised multiple methods to obtain converging evidence, there are several important caveats in the interpretation of their results:

(1) While the authors choose to use the term 'pathway' to call the inter-dependence between a set of visual regions and default-mode regions, their results have not convincingly demonstrated a definitive route of neural processing or travelling. Instead, the findings reveal a set of DMN regions are functionally more connected with object-related regions compared to place-related regions. The results are very much dependent on masking and thresholding, and the patterns can change drastically if different masks or thresholds are used.

We would like to qualify that our findings do not only reveal a set of *any* “DMN regions that are functionally more connected with object-related regions compared to place-related regions”. Instead, we show a double dissociation based on our functional task responses: DMN regions that were more responsive to semantic decisions about objects are more functionally and structurally connected to visual regions more activated by perceiving objects, while DMN regions that were more responsive to spatial decisions are more connected to visual regions activated by the contrast of scene over object perception.

We do not believe that the thresholding or masking involved in generating seeds strongly affected our results. We are reassured of this by two facts:

(1) We re-analysed the resting-state data using a stricter clustering threshold and this did not change the pattern of results (see response below).

(2) In response to a point by reviewer #2, we re-analysed the data eroding the masks of the MT-DMN, and this also didn’t change the pattern of results (please see response to reviewer 2).

In this way, our results are robust to variations in mask shape/size and thresholding.

(2) Ideally, if the authors could demonstrate the dynamics between the visual cortex and DMN in the primary task data, it would be very convincing evidence for characterising the journey from the visual cortex to DMN. Instead, the current connectivity results are derived from a separate set of resting state data. While the advantage of the authors' approach is that they are able to verify certain visual regions are more connected with certain DMN regions even under a task-free situation, it falls short of explaining how these regions dynamically interact to convert vision into semantic/spatial decision.

We agree that a valuable future direction would be to collect evidence of spatiotemporal dynamics of propagation of information along these pathways. This could be the focus of future studies designed to this aim, and we have suggested this in the manuscript based on the reviewer’s suggestion. Furthermore, as stated above, we have now qualified our use of the term ‘pathway’ in the manuscript to avoid confusion.

“These pathways refer to regions that are coupled, functionally or structurally, together, providing the potential for communication between them.”

(3) There are several results that are difficult to interpret, such as their psychophysiological interactions (PPI), representational similarity analysis, and gradient analysis. For example, typically for PPI analysis, researchers interrogate the whole brain to look for PPI connectivity. Their use of targeted ROI is unusual, and their use of spatially extensive clusters that encompass fairly large cortical zones in both occipital and temporal lobes as the PPI seeds is also an unusual approach. As for the gradient analysis, the argument that the semantic task is higher on Gradient 1 than the spatial task based on the statistics of p-value = 0.027 is not a very convincing claim (unhelpfully, the figure on the top just shows quite a few blue 'spatial dots' on the hetero-modal end which can make readers wonder if the spatial context task is really closer to the unimodal end or it is simply the authors' statistical luck that they get a p-value under 0.05). While it is statistically significant, it is weak evidence (and it is not pertinent to the main points the authors try to make).

To streamline the manuscript, we have moved the PPI and RSA results to the

Supplementary Materials. However, we believe the gradient analysis is highly pertinent to understanding the functional separation of these pathways. In the revision, we show that not only was the contrast between the Semantic and Spatial tasks significant, in addition, the majority of participants exhibited a pattern consistent with the result we report. To show the results more clearly, we have added a supplementary figure (Figure S8) focussed on comparisons at the participant level.

This figure shows the position in the gradient for each peak per participant per task. The peaks for each participant across tasks are linked with a line. Cases where the pattern was reversed are highlighted with dashed lines (7/27 participants in each gradient). This allows the reader and reviewer to verify in how many cases, at the individual level, the pattern of results reported in the text held (see “Supplementary Analysis: Individual Location of pathways in whole-brain gradients”).

(3) My suggestion for the authorsThere are several conceptual-level suggestions that I would like to offer to the authors:(1) If the pathway explanation is the key argument that you wish to convey to the readers, an effective connectivity type of analysis, such as Granger causality or dynamic caudal modelling, would be helpful in revealing there is a starting point and end point in the pathway as well as revealing the directionality of neural processing. While both of these methods have their issues (e.g., Granger causality is not suitable for haemodynamic data, DCM's selection of seeds is susceptible to bias, etc), they can help you get started to test if the path during task performance does exist. Alternatively, travelling wave type of analysis (such as the results by Raut et al. 2021 published in Science Advances) can also be useful to support your claims of the pathway.

As we have stated above, we agree with the reviewer that, given the pattern of results obtained in our work, analyses that characterise the spatiotemporal dynamics of transmission of information along the pathways would be of interest. However, we are concerned that our data is not well-optimised for these analyses.

(2) I think the thresholding for resting state data needs to be explained - by the look of Figure 2E and 3E, it looks like whole-brain un-thresholded results, and then you went on to compute the conjunction between these un-thresholded maps with network templates of the visual system and DMN. This does not seem statistically acceptable, and I wonder if the conjunction that you found would disappear and reappear if you used different thresholds. Thus, for example, if the left IFG cluster (which you have shown to be connected with the visual object regions) would disappear when you apply a conventional threshold, this means that you need to seriously consider the robustness of the pathway that you seek to claim... it may be just a wild goose that you are chasing.

We believe the reviewer might be confused regarding the procedure we followed to generate the ROIs used in the pathways connectivity analysis. As stated in the last paragraph of the “Probe phase” and “Decision phase” results subsections, the maps the reviewer is referring to (Fig. 3E, for example) were generated by seeding the intersection of our thresholded univariate analysis (Fig. 3A) with network templates. In the case of Fig 3E, these are the Semantic>Spatial decision results after thresholding, intersected with Yeo DMN (MT, FT and Core, combined). These seeds were then entered into a whole-brain seed-based spatial correlation analysis, which was thresholded and cluster-corrected using the defaults of CONN. The same is true for Fig. 2E, but using the thresholded Probe phase

Semantic>Context regions. Thus, we do not believe the objections to statistical rigour the reviewer is raising apply to our results.

The thresholding of the resting-state data itself was explained in the Methods (Spatial Maps and Seed-to-ROI Analysis). As stated above, we thresholded using the default of the CONN software package we used (cluster-forming threshold of p=.05, equivalent to T=1.65). For increased rigour, we reproduced the thresholded maps from Figs 2E and 3E further increasing the threshold from p=.05, equivalent to T=1.65, to p=.001, equivalent to T=3.1. The resulting maps were very similar, showing minimal change with a spatial correlation of r > .99 between the strict and lax threshold versions of the maps for both the probe and decision seeds. This can be seen in Figure 2E and Figure 33E, which depict the maps produced with stricter thresholding. These maps can also be downloaded from the Neurovault collection, and the re-analysis is now reported in the Supplementary Materials (see section “Supplementary Analysis: Resting-state maps with stricter thresholding”) Probe phase (compare with Fig. 2E):

(3) There are several analyses that are hard to interpret and you can consider only reporting them in the supplementary materials, such as the PPI results and representational similarity analysis, as none of these are convincing. These analyses do not seem to add much value to make your argument more convincing and may elicit more methodological critiques, such as statistical issues, the set-up of your representational theory matrix, and so on.

We have moved the PPI and RSA results to the supplementary materials. We agree this will help us streamline the manuscript.

**Reviewer #2 (Public Review):**
Summary:In this manuscript, Alam et al. sought to understand how memory interacts with incoming visual information to effectively guide human behavior by using a task that combines spatial contexts (houses) with objects of one or multiple semantic categories. Three additional datasets (all from separate participants) were also employed: one that functionally localized regions of interest (ROIs) based on subtractions of different visually presented category types (in this case, scenes, objects, and scrambled objects); another consisting of restingstate functional connectivity scans, and a section of the Human Connectome Project that employed DTI data for structural connectivity analysis. Across multiple analyses, the authors identify dissociations between regions preferentially activated during scene or object judgments, between the functional connectivity of regions demonstrating such preferences, and in the anatomical connectivity of these same regions. The authors conclude that the processing streams that take in visual information and support semantic or spatial processing are largely parallel and distinct.Strengths:(1) Recent work has reconceptualized the classic default mode network as two parallel and interdigitated systems (e.g., Braga & Buckner, 2017; DiNicola et al., 2021). The current manuscript is timely in that it attempts to describe how information is differentially processed by two streams that appear to begin in visual cortex and connect to different default subnetworks. Even at a group level where neuroanatomy is necessarily blurred across individuals, these results provide clear evidence of stimulus-based dissociation.(2) The manuscript contains a large number of analyses across multiple independent datasets. It is therefore unlikely that a single experimenter choice in any given analysis would spuriously produce the overall pattern of results reported in this work.

We thank the reviewer for their remarks on the strengths of our manuscript.

Weaknesses:(1) Throughout the manuscript, a strong distinction is drawn between semantic and spatial processing. However, given that only objects and spatial contexts were employed in the primary experiment, it is not clear that a broader conceptual distinction is warranted between "semantic" and "spatial" cognition. There are multiple grounds for concern regarding this basic premise of the manuscript.a. One can have conceptual knowledge of different types of scenes or spatial contexts. A city street will consistently differ from a beach in predictable ways, and a kitchen context provides different expectations than a living room. Such distinctions reflect semantic knowledge of scene-related concepts, but in the present work spatial and "all other" semantic information are considered and discussed as distinct and separate.

The “building” contexts we created were arbitrary, containing beds, desks and an assortment of furniture that did not reflect usual room distributions, i.e., a kitchen next to a dining room. We have made this aspect of our stimuli clearer in the Materials section of the task.

“The learning phase employed videos showing a walk-through for twelve different buildings (one per video), shot from a first-person perspective. The videos and buildings were created using an interior design program (Sweet Home 3D). Each building consisted of two rooms: a bedroom and a living room/office, with an ajar door connecting the two rooms. The order of the rooms (1st and 2nd) was counterbalanced across participants. Each room was distinctive, with different wallpaper/wall colour and furniture arrangements. The building contexts created by these rooms were arbitrary, containing furniture that did not reflect usual room distributions (i.e., a kitchen next to a dining room), to avoid engaging further conceptual knowledge about frequently-encountered spatial contexts in the real world.”

To help the reviewer and readers to verify this and come to their own conclusions, we have also added the videos watched by the participants to the OSF collection.

“A full list of pictures of the object and location stimuli employed in this task, as well as the videos watched by the participants can be consulted in the OSF collection associated with this project under the components OSF>Tasks>Training. “

We agree that scenes or spatial contexts have conceptual characteristics, and we actually manipulated conceptual information about the buildings in our task, in order to assess the neural underpinnings of this effect. In half of the buildings, the rooms/contexts were linked through the presence of items that shared a common semantic category (our “same category building” condition): this presented some conceptual scaffolding that enabled participants to link two rooms together. These buildings could then be contrasted with “mixed category buildings” where this conceptual link between rooms was not available. We found that right angular gyrus was important in the linking together of conceptual and spatial information, in the contrast of same versus mixed category buildings.

b. As a related question, are scenes uniquely different from all other types of semantic/category information? If faces were used instead of scenes, could one expect to see different regions of the visual cortex coupling with task-defined face > object ROIs? The current data do not speak to this possibility, but as written the manuscript suggests that all (non-spatial) semantic knowledge should be processed by the FT-DMN.

Thanks for raising this important point. Previous work suggests that the human visual system (and possibly the memory system, as suggested by Deen and Freiwald, 2021) is sensitive to perceptual categories important to human behaviour, including spatial, object and social information. Previous work (Silson et al., 2019; Steel et al., 2021) has shown domain-specific regions in visual regions (ventral temporal cortex; VTC) whose topological organisation is replicated in memory regions in medial parietal cortex (MPC) for faces and places. In these studies, adding objects to the analyses revealed regions sensitive to this category sandwiched between those responsive to people and places in VTC, but not in MPC. However, consistent with our work, the authors find regions sensitive to memory tasks for places and objects (as well as people) in the lateral surface of the brain.

Our study was not designed to probe every category in the human visual system, and therefore we cannot say what would happen if we contrasted social judgments about faces with semantic judgments about objects. We have added this point as a limitation and future direction for research:

“Likewise, further research should be carried out on memory-visual interactions for alternative domains. Our study focused on spatial location and semantic object processing and therefore cannot address how other categories of stimuli, such as faces, are processed by the visual-tomemory pathways that we have identified. Previous work has suggested some overlap in the neurobiological mechanisms for semantic and social processing (Andrews-Hanna et al., 2014; Andrews-Hanna & Grilli, 2021; Chiou et al., 2020), suggesting that the FT-DMN pathway may be highlighted when contrasting both social faces and semantic objects with spatial scenes. On the other hand, some researchers have argued for a ‘third pathway’ for aspects of social visual cognition (Pitcher & Ungerleider, 2021; Pitcher, 2023). Future studies that probe other categories will be able to confirm the generality (or specificity) of the pathways we described.”

c. Recent precision fMRI studies characterizing networks corresponding to the FT-DMN and MTL-DMN have associated the former with social cognition and the latter with scene construction/spatial processing (DiNicola et al., 2020; 2021; 2023). This is only briefly mentioned by the authors in the current manuscript (p. 28), and when discussed, the authors draw a distinction between semantic and social or emotional "codes" when noting that future work is necessary to support the generality of the current claims. However, if generality is a concern, then emphasizing the distinction between object-centric and spatial cognition, rather than semantic and spatial cognition, would represent a more conservative and bettersupported theoretical point in the current manuscript.

We appreciate this comment and we have spent quite a bit of time considering what the most appropriate terminology would be. The distinction between object and spatial cognition is largely appropriate to our probe phase, although we feel this label is still misleading for two reasons:

First, we used a range of items from different semantic categories, not just “objects”, although we have used that term as a shorthand to refer to the picture stimuli we presented. The stimuli include both animals (land animals, marine animals and birds) and man-made objects (tools, musical instruments and sports equipment). This category information is now more prominent in the rationale (Introduction) and the Methods to avoid confusion.

Interested readers can also review our “object” stimuli in the OSF collection associated with this manuscript:

Introduction: “…participants learned about virtual environments (buildings) populated with objects belonging to different, heterogeneous, semantic categories, both man-made (tools, musical instruments, sports equipment) and natural (land animals, marine animals, birds).”

Methods:

“A full list of pictures of the object and location stimuli employed in this task can be consulted in the OSF collection associated with this project under the components OSF>Tasks>Training.”

Secondly, we manipulated the task demands so that participants were making semantic judgments about whether two items were in the same category, or spatial judgments about whether two rooms had been presented in the same building. Our use of the terms “semantic” and “spatial” was largely guided by the tasks that participants were asked to perform.

We have revised the terminology used in the discussion to reflect this more conservative term. However, since the task performed was semantic in nature (participants had to judge whether items belonged to semantic categories), we have modified the term proposed by the reviewer to “object-centric semantics”, which we hope will avoid confusion.

(2) Both the retrosplenial/parieto-occipital sulcus and parahippocampal regions are adjacent to the visual network as defined using the Yeo et al. atlas, and spatial smoothness of the data could be impacting connectivity metrics here in a way that qualitatively differs from the (non-adjacent) FT-DMN ROIs. Although this proximity is a basic property of network locations on the cortical surface, the authors have several tools at their disposal that could be employed to help rule out this possibility. They might, for instance, reduce the smoothing in their multi-echo data, as the current 5 mm kernel is larger than the kernel used in Experiment 2's single-echo resting-state data. Spatial smoothing is less necessary in multiecho data, as thermal noise can be attenuated by averaging over time (echoes) instead of space (see Gonzalez-Castillo et al., 2016 for discussion). Some multi-echo users have eschewed explicit spatial smoothing entirely (e.g., Ramot et al., 2021), just as the authors of the current paper did for their RSA analysis. Less smoothing of E1 data, combined with a local erosion of either the MTL-DMN and VIS masks (or both) near their points of overlap in the RSFC data, would improve confidence that the current results are not driven, at least in part, by spatial mixing of otherwise distinct network signals.

A: The proximity of visual peripheral and DMN-C networks is a property of these networks’ organisation (Silson et al., 2019; Steel et al., 2021), and we agree the potential for spatial mixing of the signal due to this adjacency is a valid concern. Altering the smoothing kernel of the multi-echo data would not address this issue though, since no connectivity analyses were performed in task data. The reviewer is right about the kernel size for task data (5mm), but not about the single echo RS data, which actually has lower spatial resolution (6mm).

Since this objection is largely about the connectivity analysis, we re-analysed the RS data by shrinking the size of the visual probe and DMN decision ROIs for the context task using fslmaths. We eroded the masks until the smallest gap between them exceeded the size of our 6mm FWHM smoothing kernel, which eliminates the potential for spatial mixing of signals due to ROI adjacency. The eroded ROIs can be consulted in the OSF collection associated with this project see component “ROI Analysis/Revision_ErodedMasks”. The results, presented in the supplementary materials as “Eroded masks replication analysis”, confirmed the pattern of findings reported in the manuscript (see SM analysis below). We did not erode the respective ROIs for the semantic task, given that adjacency is not an issue there.

“Eroded masks replication analysis:

The Visual-to-DMN ANOVA showed main effects of seed (F(1,190)=22.82, p<.001), ROI (F(1,190)=9.48, p=.002) and a seed by ROI interaction (F(1,190)=67.02, p<.001). Post-hoc contrasts confirmed there was stronger connectivity between object probe regions and semantic versus spatial context decision regions (t(190)=3.38, p<.001), and between scene probe regions and spatial context versus semantic decision regions (t(190)=-7.66, p<.001).

The DMN-to-Visual ANOVA confirmed this pattern: again, there was a main effect of ROI (F(1,190)=4.3, p=.039) and a seed by ROI interaction (F(1,190)=57.59, p<.001), with posthoc contrasts confirming stronger intrinsic connectivity between DMN regions implicated in semantic decisions and object probe regions (t(190)=5.06, p<.001), and between DMN regions engaged by spatial context decisions and scene probe regions (t(190)=3.25, p=.001).”

(3) The authors identify a region of the right angular gyrus as demonstrating a "potential role in integrating the visual-to-DMN pathways." This would seem to imply that lesion damage to right AG should produce difficulties in integrating "semantic" and "spatial" knowledge. Are the authors aware of such a literature? If so, this would be an important point to make in the manuscript as it would tie in yet another independent source of information relevant to the framework being presented. The closest of which I am aware involves deficits in cued recall performance when associates consisted of auditory-visual pairings (Ben-Zvi et al., 2015), but that form of multi-modal pairing is distinct from the "spatial-semantic" integration forwarded in the current manuscript.

This is a very interesting observation. There is a body of literature pointing to AG (more often left than right) as an integrator of multimodal information: It has been shown to integrate semantic and episodic memory, contextual information and cross-modality content.

The Contextual Integration Model (Ramanan et al., 2017) proposes that AG plays a crucial role in multimodal integration to build context. Within this model, information that is essential for the representation of rich, detailed recollection and construction (like *who, when,* and, crucially for our findings, *what and where*) is processed elsewhere, but integrated and represented in the AG. In line with this view, Bonnici et al (2016) found AG engagement during retrieval of multimodal episodic memories, and that multivariate classifiers could differentiate multimodal memories in AG, while unimodal memories were represented in their respective sensory areas only. Recent work examining semantic processing in temporallyextended narratives using multivariate approaches concurs with a key role of left AG in context integration (Branzi et al., 2020).

In addition to context integration, other lines of work suggest a role of AG as an integrator across modalities, more specifically. Recent perspectives suggest a role of AG as a dynamic buffer that allows combining distinct forms of information into multimodal representations (Humphreys et al., 2021), which is consistent with the result in our study of a region that brings together semantic and spatial representations in line with task demands. Others have proposed a role of the AG as a central connector hub that links three semantic subsystems, including multimodal experiential representation (Xu et al., 2017). Causal evidence of the role of AG in integrating multimodal features has been provided by Yazar et al (2017), who studied participants performing memory judgements of visual objects embedded in scenes, where the name of the object was presented auditorily. TMS to AG impaired participants’ ability to retrieve context features across multiple modalities. However, these studies do not single out specifically right AG.

Some recent proposals suggest a causal role of right AG as a key region in the early definition of a context for the purpose of sensemaking, for which integrating semantic information with many other modalities, including vision, may be a crucial part (Seghier, 2023). TMS studies suggest a causal role for the right AG in visual attention across space

(Olk et al. 2015, Petitet et al. 2015), including visual search and the binding of stimulus- and response-characteristics that can optimise it (Bocca et al. 2015). TMS over the right AG disrupts the ability to search for a target defined by a conjunction of features (Muggleton et al. 2008) and affects decision-making when visuospatial attention is required (Studer et al. 2014). This suggests that the AG might contribute to perceptual decision-making by guiding attention to relevant information in the visual environment (Studer et al. 2014). These, taken together, suggest a causal role of right AG in controlling attention across space and integrating content across modalities in order to search for relevant information.

Most of this body of research points to left, rather than right, AG as a key region for integration, but we found regions of right AG to be important when semantic and spatial information could be integrated. We might have observed involvement of the right AG in our study, as opposed to the more-often reported left, given that people have to integrate semantic information with spatial context, which relies heavily on visuospatial processes predominantly located in right hemisphere regions (cf. Sormaz et al., 2017), which might be more strongly connected to right than left AG.

Lastly, we are not aware of a literature on right AG lesions impairing the integration of semantic and spatial information but, in the face of our findings, this might be a promising new direction. We have added as a recommendation that patients with damage to right AG should be examined with specific tasks aimed at probing this type of integration. We have added the following to the discussion:

“We found a region of the right AG that was potentially important for integrating semantic and spatial context information. Previous research has established a key role of the AG in context integration (Ramanan et al., 2017; Bonnici et al., 2016; Branzi et al., 2020) and specifically, in guiding multimodal decisions and behaviour (Humphreys et al., 2021; Xu et al., 2017; Yazar et al., 2017). Although some recent proposals suggest a causal role of right AG in the early establishment of meaningful contexts, allowing semantic integration across modalities (Seghier, 2023; Olk et al., 2015, Petitet et al., 2015; Bocca et al., 2015; Muggleton et al. 2008), the majority of this research points to left, rather than right, AG as a key region for integration. However, we might have observed involvement of the right AG in our study given that people were integrating semantic information with spatial context, and right-lateralised visuospatial processes (cf. Sormaz et al., 2017) might be more strongly connected to right than left AG. We are not aware of a literature on right AG lesions impairing the integration of semantic and spatial information but, in the face of our findings, this might be a promising new direction. Patients with damage to right AG should be examined with specific tasks aimed at probing this type of integration.”

**Recommendations for the authors:**

**Reviewer #2 (Recommendations For The Authors):**
(1) I mentioned the numerous converging analyses reported in this manuscript as a strength. However, in practice, it also makes results in numerous dense figures (routinely hitting 7-8 sub-panels) and results paragraphs which, as currently presented, are internally coherent but are not assembled into a "bigger picture" until the discussion. Readers may have an easier time with the paper if introductions to the different analyses ("probe phase", "decision phase", etc.) also include a bigger-picture summary of how the specific analysis is contributing to the larger argument that is being constructed throughout the manuscript. This may also help readers to understand why so many different analysis approaches and decisions were employed throughout the manuscript, why so many different masks were used, etc.

Thank you for this suggestion. We agree that the range of analyses and their presentation can make digesting them difficult. To address this, we have outlined our analyses rationale at the beginning of the results as a sort of “big picture” summary that links all analyses together, and added introductory paragraphs to each analysis that needed them (namely, the probe, decision, and pathway connectivity analyses, as the gradient and integration analyses already had introductory paragraphs describing their rationale, and the PPI/RSA analyses were moved to supplementary materials), linking them to the summary, which we reproduce below:

“To probe the organisation of streams of information between visual cortex and DMN, our neuroimaging analysis strategy consisted of a combination of task-based and connectivity approaches. We first delineated the regions in visual cortex that are engaged by the viewing of probes during our task (Figure 2), as well as the DMN regions that respond when making decisions about those probes (Figure 3): we characterised both by comparing the activation maps with well-established DMN and object/scene perception regions, analysed the pattern of activation within them, their functional connectivity and task associations. Having characterised the two ends of the stream, we proceeded to ask whether they are differentially linked: are the regions activated by object probe perception more strongly linked to DMN regions that are activated when making semantic decisions about object probes, relative to other DMN regions? Is the same true for the spatial context probe and decision regions? We answered this question through a series of connectivity analyses (Figure 4) that examined: (1) if the functional connectivity of visual-to-DMN regions (and DMN-to-visual regions) showed a dissociation, suggesting there are object semantic and spatial cognition processing ‘pathways’; (2) if this pattern was replicated in structural connectivity; (3) if it was present at the level of individual participants, and, (4) we characterised the spatial layout, network composition (using influential RS networks) and cognitive decoding of these pathways. Having found dissociable pathways for semantic (object) and spatial context (scene) processing, we then examined their position in a high-dimensional connectivity space (Figure 5) that allowed us to document that the semantic pathway is less reliant on unimodal regions (i.e., more abstract) while the spatial context pathway is more allied to the visual system. Finally, we used uni- and multivariate approaches to examine how integration between these pathways takes place when semantic and spatial information is aligned (Figure 6).”

(2) At various points, figures are arranged out of sequence (e.g., panel d is referenced after panel g in Figure 2) or are missing descriptions of what certain colors mean (e.g., what yellow represents in Figure 6d). This is a minor issue, but one that's important and easy to address in future revisions.

We thank the reviewer for bringing this issue to our attention. We have added descriptions for the yellow colour to the figure legends of Figures 6 and 7 (now in supplementary materials, Figure S9).

We have also edited the text to follow a logical sequence with respect to referencing the panels in Figures 2 and 3, where panel d is now referenced after panel c. Lastly, we reorganised the layout of Figure 4 to follow the description of the results in the text.